# Transcriptional drifts associated with environmental changes in endothelial cells

**Yalda Afshar[1,2], Feyiang Ma[2,3], Austin Quach[3], Anhyo Jeong[1], Hannah L Sunshine[4,5], Vanessa Freitas[6], Yasaman Jami-Alahmadi[7], Raphael Helaers[8], Xinmin Li[9], Matteo Pellegrini[2,3], James A Wohlschlegel[7], Casey E Romanoski[10], Miikka Vikkula[8,11], M Luisa Iruela-Arispe[3,5]***

[1]Department of Obstetrics and Gynecology, University of California, Los Angeles, Los Angeles, United States; [2]Molecular Biology Institute, University of California, Los Angeles, Los Angeles, United States; [3]Department of Molecular, Cell, and Developmental Biology, University of California, Los Angeles, Los Angeles, United States; [4]Department of Molecular, Cellular and Integrative Physiology, University of California, Los Angeles, Los Angeles, United States; [5]Department of Cell and Developmental Biology, Northwestern University Feinberg School of Medicine, Chicago, United States; [6]Departament of Cell and Developmental Biology, Institute of Biomedical Science, University of Sao Paulo, Los Angeles, United States; [7]Department of Biological Chemistry, University of California, Los Angeles, United States; [8]Human Molecular Genetics, de Duve Institute, University of Louvain, Brussels, Belgium; [9]Department of Pathology and Laboratory Medicine, University of California, Los Angeles, United States; [10]Department of Cellular and Molecular Medicine, University of Arizona, Tucson, United States; [11]WELBIO department, WEL Research Institute, Wavre, Belgium

*For correspondence:
arispe@northwestern.edu

**Competing interest:** The authors declare that no competing interests exist.

**Abstract** Environmental cues, such as physical forces and heterotypic cell interactions play a critical role in cell function, yet their collective contributions to transcriptional changes are unclear. Focusing on human endothelial cells, we performed broad individual sample analysis to identify transcriptional drifts associated with environmental changes that were independent of genetic background. Global gene expression profiling by RNA sequencing and protein expression by liquid chromatography–mass spectrometry directed proteomics distinguished endothelial cells in vivo from genetically matched culture (in vitro) samples. Over 43% of the transcriptome was significantly changed by the in vitro environment. Subjecting cultured cells to long-term shear stress significantly rescued the expression of approximately 17% of genes. Inclusion of heterotypic interactions by co-culture of endothelial cells with smooth muscle cells normalized approximately 9% of the original in vivo signature. We also identified novel flow dependent genes, as well as genes that necessitate heterotypic cell interactions to mimic the in vivo transcriptome. Our findings highlight specific genes and pathways that rely on contextual information for adequate expression from those that are agnostic of such environmental cues.

## Editor's evaluation

The findings of this study, focused on endothelial cells, are fundamental and of broad interest to various fields using cultured primary cells. The authors provide compelling evidence of how the culture conditions impact on gene expression.

## Introduction

Endothelial cells define the functional integrity and response to hemodynamic blood forces on the luminal surface of blood vessels (*Iruela-Arispe and Davis, 2009*). They are also responsible for the selective trafficking of immune cells, regulation of metabolites and fluid extravasation to tissues (*Jackson, 2019*; *Sun and Feinberg, 2015*; *Vandenbroucke et al., 2008*; *Wettschureck et al., 2019*). More recently, it has become clear that the endothelium provides instructive angiocrine signals required for the differentiation of tissues during development and for homeostasis of organs in the adult (*Gomez-Salinero and Rafii, 2018*). In fact, it is challenging to identify a single pathological condition that could not be either worsened or improved by affecting the biology of blood vessels. Either through regulation of barrier function, anti-thrombotic properties, angiocrine or angiogenic capacity, endothelial cells have broad impact and therapeutic reach. Thus, there is a compelling incentive to define the mechanisms that control endothelial function and explore strategies to alter these functions as we work toward understanding disease etiology and processes leading to restore normal organ physiology.

Much of the knowledge accumulated on endothelial cell function has emerged through studies in vitro. The ability to grow endothelial cells under culture conditions has enabled investigators to identify growth factors that promote endothelial growth (*Apte et al., 2019*; *Gerber et al., 1998*), define the molecules involved in barrier function (*Christensen et al., 2016*; *Corada et al., 2019*; *Trani and Dejana, 2015*), and recognize discrete steps in leukocyte–endothelial selection and extravasation (*Muller, 2016*). However, a complete reductionist in vitro (culture) approach deprives endothelial cells from contextual information which could impact experimental read-outs.

As for all cells, the endothelial cell transcriptome is dependent on their native environmental milieu which includes homo- and heterotypic cell interactions, soluble factors, three-dimensional organization (*Wang et al., 2022*), and physical forces (*Choi and Seo, 2019*; *Cleuren et al., 2019*; *Dayang et al., 2019*; *Jambusaria et al., 2020*).This contextual information is removed when cells are placed in vitro. While endothelial identity and many biological aspects are retained, there is no frame of reference, meaning comparison to in vivo state, to determine what has been lost, and what could have been artificially gained, during the transition to an in vitro environment. Such gains and losses are likely to affect conclusions drawn from in vitro expression profiles. Yet, without an understanding of these changes, the validity of conclusions associated with experimental challenge remains uncertain.

To gain more clarity on the impact of culture conditions on endothelial cells, we set out to evaluate human umbilical vein endothelial cells (HUVECs) directly upon removal from the cord (in vivo/cord) and after exposing the same cells to short- and long-term in vitro culture. After defining the gene signatures changed in culture, we inquired as to whether in vitro environmental exposure to shear stress and interactions with smooth muscle cells (SMCs) were able to ameliorate the differential expression signatures and 'correct' and 'rescue' drifts. Through this process and relying on genetically identical in vivo transcriptome, we identified groups of genes exquisitely dependent on long-term shear stress and others dependent on heterotypic cell interactions. Importantly, we also identified a large cohort of genes that were unable to regain levels comparable to in vivo settings and others that were artificially induced by exposure to culture conditions. Together, this work has implications for enabling investigations of endothelial cells with improved fidelity to in vivo phenotypes that should improve reproducibility and translation of experimental findings.

## Results

### Transcriptional drifts associated with the transition of in vivo (cord) to in vitro (culture)

To uncover changes on endothelial cells as result of exposure to culture conditions, we evaluated the transcriptome of endothelial cells isolated from human umbilical cord veins. Half of each participant's cell preparation was freshly processed for RNA isolation (referred to as 'cord' or 'in vivo') while the other half was placed under culture conditions (referred to as 'culture' or 'in vitro'). Cells were subsequently passaged and evaluated at 'early' passage (P 2–3) and 'late' passage (P 7–8) to capture transcriptional differences between cellular environments that were common amongst all seven patients regardless of fetal sex or genetic background (*Figure 1A*, *Supplementary file 1*). Patient demographics with paired maternal–fetal outcomes are provided in *Table 1*, and each patient had matched

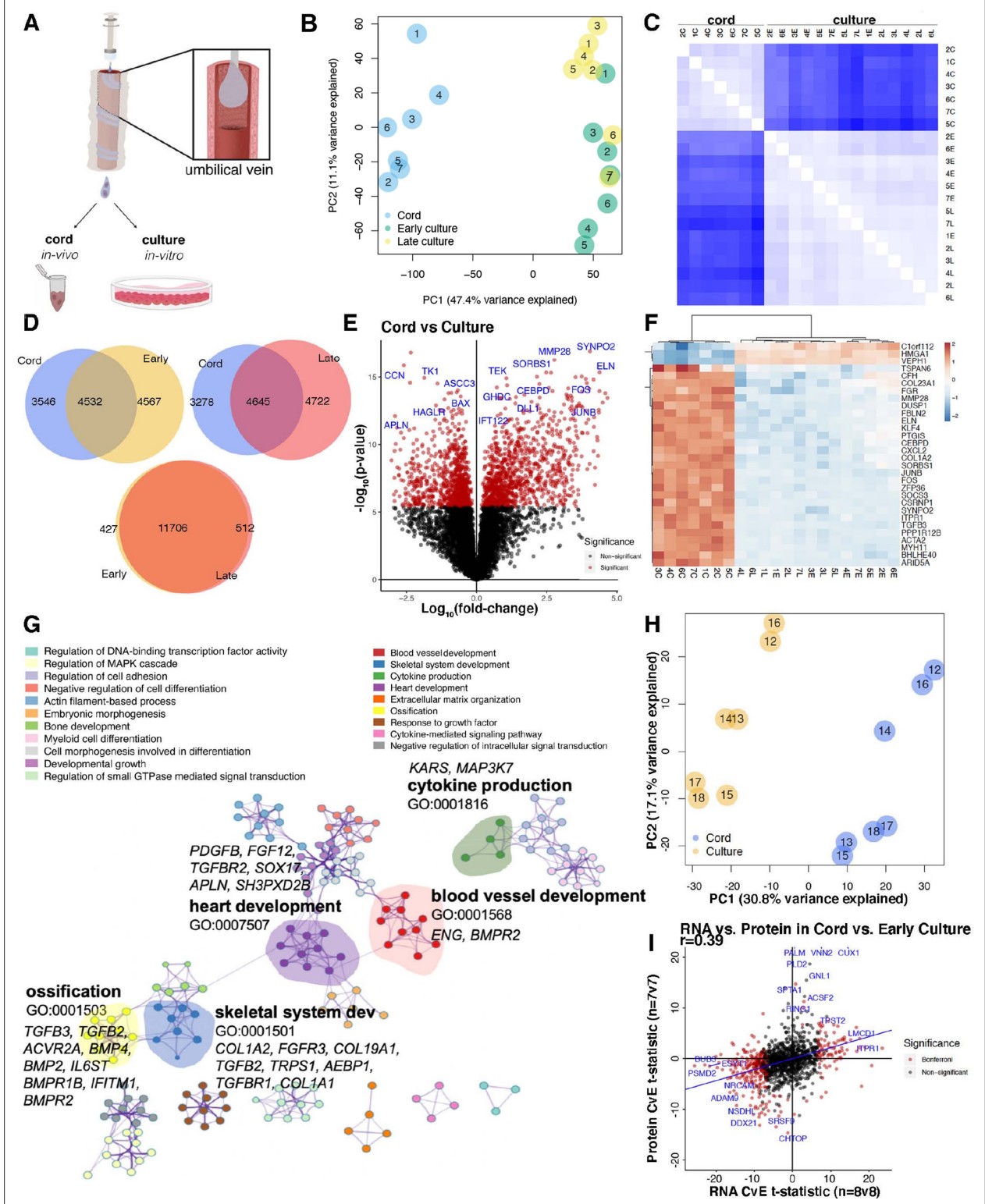

**Figure 1.** Human umbilical cord endothelial cell transcriptome. (**A**) Model of endothelial cell collection *for* in vivo (cord) and in vitro (culture) experiments. Endothelial cells are isolated in a slurry and used immediately for downstream experiments or cultured for subsequent passages. (**B**) Principal component analysis (PCA) of transcriptome of the seven matched cord, early culture, and late culture samples with significant separation along PC1. (**C**) Spearman correlation demonstrating inter-condition (cord = C, early passage = E, late passage = L) and intra-sample variability with *k*-means clustering by cord. (**D**) 40–45% of the of the expressed genes overlapped in relative expression patterns between cord and culture regardless of early and late cultures. While early and late cultures overlap in 93% of the genes. (**E**) Volcano plot of genes most significantly expressed in cord (right) versus

*Figure 1 continued on next page*

*Figure 1 continued*

culture (left) by log10 fold change. (**F**) Heatmap of top 30 differentially expressed genes in 21 samples from 7 individuals expressed between cord and culture. (**G**) Network profile of subset of Gene Expression Omnibus (GEOs) significant in cord versus culture. GEO is represented by cluster identity and each term is represented as circle node visualized on Metascape. The highlighted GEOs are the most significant pathways by p value. (**H**) Mass spectrometry proteomic profile of seven matched cord and culture separated by cord and culture on PC1. (**I**) Scatter plot depicting RNA *t*-statistics (cord/culture) versus protein *t*-statistics (cord/culture) with a correlation coefficient of *r* = 0.4.

The online version of this article includes the following figure supplement(s) for figure 1:

**Figure supplement 1.** Endothelial cell isolation.

**Figure supplement 2.** RNAseq cord versus culture.

cord, early passage, and late passage paradigm. Principal component analysis (PCA) of bulk RNA sequencing (RNAseq) transcriptional profiles revealed that cord versus culture/in vitro environments were the dominant factor influencing measured expression levels (*Figure 1B, C*, *Supplementary file 1*). PC1 captured 47.4% of the total variance whereas PC2 only accounts for 11.1%. PC2 appeared to represent the differences between early and late passage but these conditions did not segregate from each other as clearly as cord versus in vitro culture. As such, all comparisons to culture conditions are subsequently made with early (in vitro) cells in culture.

Approximately half of the expressed genes were differentially expressed between cord and culture conditions (4532–4645 genes overlapping), whereas the transcriptomic signature was very similar

**Table 1.** Patient demographics.

|  | Sample code | Gestational age | Fetal sex | Race |
|---|---|---|---|---|
| **RNAseq** | | | | |
| | 1 | 40w1d | M | Asian, Vietnamese |
| | 2 | 39w4d | F | Asian, Chinese |
| | 3 | 39w4d | M | Asian |
| | 4 | 39w1d | M | Asian |
| | 5 | 39w0d | F | White |
| Cord, early culture p2–3, and late culture p7–8 experiments | 6 | 37w5d | M | White |
| | 7 | 38w4d | F | White |
| **Flow, RNAseq** | | | | |
| | 8 | 39w6d | F | White |
| | 9 | 40w5d | F | Black |
| | 10 | 40w4d | F | Asian, Chinese |
| Culture static versus culture flow experiments | 11 | 39w5d | M | White |
| **Proteomics** | | | | |
| | 12 | 40w2d | M | Asian, Indian |
| | 13 | 39w3d | F | Latino |
| | 14 | 37w2d | M | Latino |
| | 15 | 39w3d | M | Asian, Chinese |
| | 16 | 38w5d | F | Latino |
| | 17 | 37w0d | M | Asian, Other |
| Cord versus culture experiments | 18 | 40w0d | F | White |
| **scRNAseq** | | | | |
| Culture (monoculture) versus co-culture experiments | SMC | 37w3d | F | Latino (Other) |
| | EC | 36w4d | M | Other |

between early and late cultures (11,706 genes overlapping) (*Figure 1D*). As such, we considered only differences between cord and culture signatures going forward (*Figure 1E–G*, *Supplementary file 1*, *Figure 1—figure supplements 1–2*).

Genes with robust changes in expression are highlighted in *Figure 1E, F*. Among several signatures, we observed that TGFβ and BMP target genes were reduced under culture. Some of the most in vivo-specific transcripts were related to the extracellular matrix; while several genes specific to the in vitro environment associated with the cell cycle (*Supplementary file 1*). We also found that the most highly expressed genes across patients and environments demonstrated minimal variation across individuals and considerable variation between environments (*Figure 1F*, *Supplementary file 1*). As expected, we found that endothelial cells lose expression of flow-responsive genes (*KLF4*, *KLF2*) once placed under culture conditions, whereas they quickly acquire proliferation-related genes (*CCNB2*, *CCNA2*, *CDCA2*). Perhaps more surprising was that transition into culture promotes a significant decrease in transcripts associated with extracellular matrix genes (*COL23A1*, *MMP28*, *FBLN2*, *ELN*, *COL1A2*, *COL6A3*), cytokine (*CXCL2*, *SOCS3*, *TGFB3*, *CTGF*), and early response genes (*FOS*, *ZFP36*, *JUNB*) (*Supplementary file 1*, *Supplementary file 4*). In addition to increased expression of cell cycle genes in culture, transcripts associated with survival and a pro-angiogenic phenotype were also upregulated (e.g., *APLN*, *BAX*, *CCN*, *CCNB2*, *CCNBA1*, *CEPH1*, *CDCA7l*, *CDCA2*, *MDM2*). Further, the significant increase of VEPH1 under culture conditions was of particular interest as the protein product of this gene is associated with suppression of TGFβ1, FOXO, and Wnt signaling (*Shathasivam et al., 2015*).

Gene Ontology (GO) term enrichment of differentially expressed genes was performed using GO biological processes in order of significance. Significant terms, defined using hypergeometric p values and enrichment factors, were hierarchically clustered based on similarities among gene members into networks (*Figure 1G*).

In the network, terms are represented by a node with its size proportional to the number of differentially expressed genes in that term. Focusing on genes expressed uniquely in cord relative to culture, we found enrichment of transcripts with documented involvement into blood vessel development, skeletal system development (mostly the TGFβ family), heart/blood vessel development, ossification (extracellular matrix genes), and cytokine production (*Figure 1G*).

To determine whether the identified changes were supported by similar drifts at the protein level, validation of the transcriptomic signature was performed by comparing cord and in vitro protein extracts by untargeted liquid chromatography–mass spectrometry (LC–MS)-based proteomics. PCA analysis of relative protein abundances was conducted for seven matched individuals from the cord and early culture. The analysis demonstrated clear separation of the experimental conditions (*Figure 1H*). In agreement to the RNA-level differences, there were significant changes in protein expression between the environments. Albeit not as remarkably different than the transcriptomic read-outs (likely due to depth of coverage and statistical power), we identified an -omics signature of proteins specific to cord (about 160/3000 proteins) and to early culture (about 411/3000 proteins) (*Figure 1—figure supplement 2E*). These differences are presented in *Figure 1—figure supplement 2F*.

To explore the degree of overlap between RNA and protein, we compared the results of differential transcripts using the cord versus culture analysis to that of protein cord versus culture analysis (*Figure 1—figure supplement 2G*). The relationships between *t*-statistics between cord and culture across -omic layers revealed significant correlation between RNA and protein signatures ($r = 0.4$, $p = 1 \times 10^{-07}$) (*Figure 1I*). Consistent with prior findings, the data revealed low expression of cell cycle proteins and high expression of flow-responsive proteins in the cord (ex vivo) proteomics profile (*Figure 1I*, *Supplementary files 2 and 9*).

## Global transcriptional changes affected by shear stress

A large number of genes associated with transition from cord to culture appeared to be flow related, we explored the potential to ameliorate these differences by imposing shear stress on cultured cells. This approach is warranted by observations that once placed under laminar flow, endothelial cells significantly change their morphology and reduce proliferation resembling in vivo conditions (*Chiu and Chien, 2011*). Further, the onset of flow is associated with significant transcriptional increase in flow-responsive genes, like *KLF2* (*Chien, 2007*; *Nakajima and Mochizuki, 2017*; *Zhou et al., 2014*), which is one of the cord-specific transcripts (*Figures 1 and 2*, *Supplementary file 3*). We thus

performed two comparisons: (1) static culture to flow cultures (*Figure 2A*) and (2) each to cord endothelial cells (*Figures 2 and 3*, *Supplementary file 3*).

Indeed, we found that shear stress significantly rescued the expression of approximately 17% of genes including targets of BMP and Notch signaling known to be sensitive to flow. At the transcriptional level, the effect of physical forces, particularly laminar, oscillatory, and disturbed flow has been extensively investigated (*Kim et al., 2017*; *Nakajima et al., 2017*; *Peng et al., 2019*; *Polacheck et al., 2017*). These investigations have been instrumental to clarify the effect of shear stress on endothelial cells. Nonetheless, previously published studies focused in evaluating flow responses at short and long time points in relation to static cultures. We took advantage of having isolated RNA directly from cords, allowing for comparisons between in vivo and in vitro (static and flow) conditions using genetically identical backgrounds.

First, PCA of matched patients ($n = 4$, *Table 1*) demonstrated static cells in vitro and under 30 min of flow displayed relatively similar global transcriptional signatures. Differences were apparent on PC1 with flow, defined as 8–48 hr of laminar shear stress exposure (*Figure 2B*). *Figure 2C* provides a clear delineation and transcriptomic signature as a function of static (control and 30 min) versus longer time points (8, 24, and 48 hr of flow). Significant changes (log10FC) were noted between static and flow cultures (*Figure 2D* and *Supplementary files 3 and 4*), with *IGFBP5*, *ELN*, *KLF4*, *ETPR1*, and *TGFBR3* significantly dependent on flow for their transcriptional increase. Correlation scatter plots of the cord versus culture (*x*-axis) were compared to time under flow (*y*-axis) and this analysis showed a time-dependent positive correlation to the cord transcriptome (vs. culture) (*Figure 2E*). Progressive time under flow from up to 48 hr of shear stress (flow) revealed that the transcriptional signature of cells correlates more specifically to that of the cord than with static cultures. Initially, the correlation coefficient was insignificant ($r = -0.035$, p = 0.004) with progressive changes to the point that by 48 hr of shear stress the correlational coefficient to cord reached $r = 0.34$, $= -8.0 \times 10^{-9}$, which is significantly different than static culture (*Figure 2E*). Collectively these data offer proof that drifts in the transcriptome of endothelial cells under culture can be partially rescued by exposure to laminar shear stress. To improve the data accessibility and analysis capabilities of the generated transcriptomic flow data, we implemented an open-source website, *Flow Profiler*, to display the data as a table, plots, and with some analysis functionalities (*Figure 2—figure supplement 2*).

A marked change toward the cord state was noted also by pathway analysis. Specifically, GO terms associated with blood vessel development (*EDN1*, *BMP2*, *BMP4*, *TGFPR3*, *ITG1BP1*, *HES1*, *HEY1*), regulation of cellular protein location (*ITGA3*, *RACK1*, *PTPN9*, *SPTBN1*), and cellular response to laminar fluid shear stress (*ASS1*, *KLF2*, *KLF4*, *MAPK7*, *NFE2L2*) were regained by long-term exposure to flow (*Figure 2F*). Gene set enrichment analysis of differential expressed genes in cultured endothelial cells under flow (vs. static) revealed gene annotations related to an acute inflammatory response, heart morphogenesis, second messenger-mediated signaling, and ossification (related to BMP and TGFβ responses). We also found tRNA and rRNA metabolic processes were silenced under flow (*Supplementary files 3 and 4*, *Figure 2—figure supplement 1*, *Figure 3—figure supplement 1*).

## Imposing shear stress in vitro partially rescues the in vivo signature

Superimposing the cord and culture signatures (from *Figure 1*) with the static versus flow experiments (from *Figure 2*) clarifies how the transcriptional profile of cultured cells under flow approximates the in vivo transcriptome better when compared to static states (*Figure 3A*, across PC1, *Supplementary files 3 and 4*). PC1 primarily displays the differences between cord and culture samples (we interpret the PC2 to represent differences between short-term and extended flow). Since the extended flow samples are in the middle position between cord and late culture, we interpret this as a partial rescue of the differences imparted by culture. This shift was also noted by evaluating total number of transcriptional changes up- or downregulated (*Figure 3B*). In fact, much of the cord signature overlapped with genes that were rescued or attenuated under flow and paralleled those expressed by the cord. Specifically, the incorporation of shear stress to the in vitro static conditions attenuated the variability between cord and culture with a drift recovery in 17% of the genes (*Figure 3*).

To identify cohorts of genes altered by shear stress and that approximate the in vivo environment (rescued), we performed a transcriptome-wide weighted gene co-expression network analysis (WGCNA). This approach led us to identify 36 co-expression modules, revealing gene groups that are co-enriched in either cord or culture environments, or in static versus flow conditions (*Figure 3C*, red:

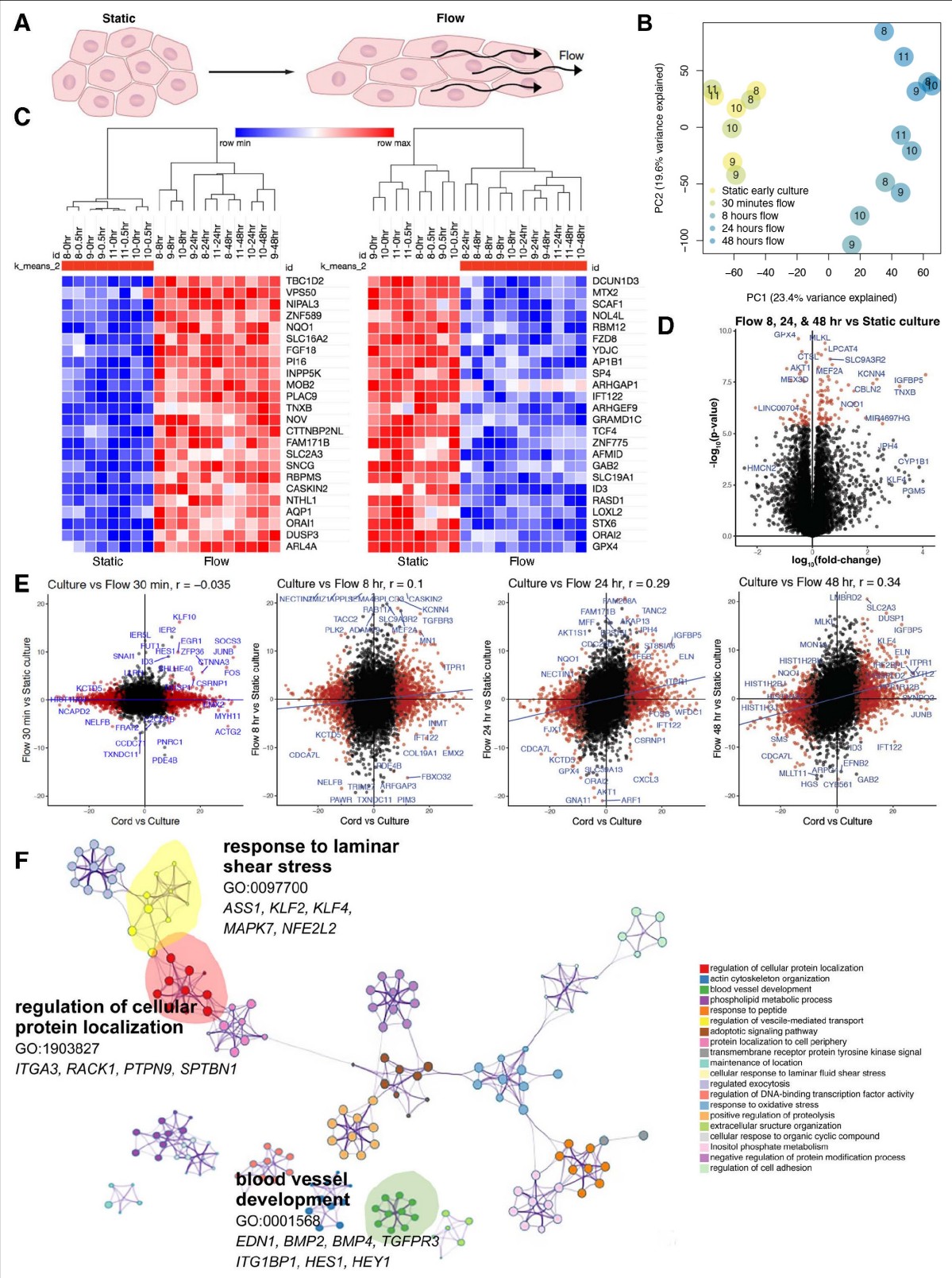

**Figure 2.** Shear stress induces a time-dependent transcriptomic flow signature. (**A**) Phenotype of in vitro flow model induces endothelial cellular shape changes under flow. (**B**) Principal component analysis (PCA) of each sample under static (yellow) and flow-conditioned endothelial cells by bulk RNA sequencing (RNAseq; blue). (**C**) Heatmap of differentially expressed genes (DEGs) by bulk RNAseq in static and flow-conditioned cells. Row z-score reflects the gene expression change. (**D**) Volcano plot illustrating statistical significance versus fold change between flow and static cultures

*Figure 2 continued on next page*

*Figure 2 continued*

demonstrating the most significantly differentially expressed genes. (**E**) Time-dependent volcano plot and correlation coefficient highlighting the correlation of flow time to cord transcriptome where longer flow correlates more strongly to cord (in vivo) transcriptome. (**F**) Network profile of subset of GEOs significant in flow versus static culture. GEO is represented by cluster identity and each term is represented as circle node visualized with Metascape.

The online version of this article includes the following source data and figure supplement(s) for figure 2:

**Figure supplement 1.** RNAseq flow versus no-flow.

**Figure supplement 2—source data 1.** Flow profiler application.

**Figure supplement 2.** Operation manual and explanatory information for the Flow Profiler.

down; blue: up, *Supplementary files 3 and 4*, *Figure 3—figure supplement 1*). Relative expression values for the most expressed modules across each of the patients are illustrated in *Figure 3—figure supplement 1D, E* and then culminated in summary in *Figure 3D*.

Superimposing the cord transcriptome on the flow transcriptome, highlighted co-expressed modules with significant enriched directionality (concordance of up- or downregulation) in cord and culture transcriptomes (*Figure 3C*). Specifically, we compared the differential gene expression of the cord to that of the cultured cells under flow. Using the differential gene expression, GO network analysis of WGCNA demonstrated that differential modules were selectively increased (blue) and decreased (brown) by long-term exposure to shear stress (*Figure 3E–H*). The blue module (2185 genes, $r = 0.71$, $p = 3e-04$) was increased in the cord and under flow conditions as compared to static culture. The blue module showed increased transcriptional concurrence with the cord and this was progressive with time under flow ($r = 0.8$, $p = 4e-04$). Although exposure to shear stress partially recapitulated the cord environment (*Figure 3E*), this was not the case for all the transcripts, highlighting signatures that are exquisitely flow dependent and others that are flow independent and likely regulated by alternative factors, such as heterotypic cell interactions or in vivo metabolites. Notably, the genes and GOs associated with this module included blood vessel development and leukocyte activation (*Figure 3F*). The brown module was decreased both in the cord and in flow as compared to static cells (1408 genes, $r = -0.9$, $p = 3e-08$). The brown module was defined by cell cycle and cell cycle checkpoints, was less expressed in cord (vs. culture) and in flow (vs. static, $r = -0.62$, $p = 0.01$). These genes gained expression in culture, yet flow reverted their phenotype to lower expression, as was evident in cord (*Figure 3G, H*, *Table 2*, *Supplementary file 5*). In summary, this network analysis uncovered co-expressed gene signatures that are sensitive to shear stress (induced, aka blue module and repressed, aka brown module) and represented in vivo.

In addition, the robust dataset identified transcripts previously unknown to be altered by laminar shear stress (*Supplementary file 3*). To confirm the reproducibility of a few transcripts at the protein level, we validated by western blot three examples found to be upregulated, downregulated, or unchanged that were further in relation to their levels in vivo (endothelial lysates from cord umbilical vein) (*Figure 4*, *Figure 4—source data 1*). Detailed evaluation of the time kinetics for transcripts uncovered important nuances, for example, the extended time course (48 hr) of shear stress is important for some transcripts. For example, thioredoxin-interacting protein (TXNIP), a stress-responsive protein that inhibits thioredoxin and previously thought to be reduced by exposure to 24 hr of shear stress [PMID:15696199], exhibits a drastic increased by 48 hr of laminar flow. A list of the top 30 transcripts that are altered from in vivo (cord) to in vitro is shown in *Table 2*. *Table 3* shows the extent of concordant and discordant rescue by exposing cells to laminar shear stress. In addition introduces a platform 'Flow Profiler' to interrogate the behavior of any gene under flow.

## Co-culture with SMCs further rescues the in vivo transcriptional profile of endothelial cells

Given these global differences between cord and culture, we asked whether the differential gene expression was also affected by heterotypic cell interactions, namely with SMCs. To address this question, we leveraged single-cell RNA sequencing (scRNAseq) technology to obtain transcriptomes of individual cells isolated from endothelial cells in a homogenous culture (mono-culture, MC) versus endothelial cells co-cultured with SMCs (co-culture, CC). The approach was aimed at further approximating contextual environment and obtain signatures responsive to those changes (*Figure 5A*). We

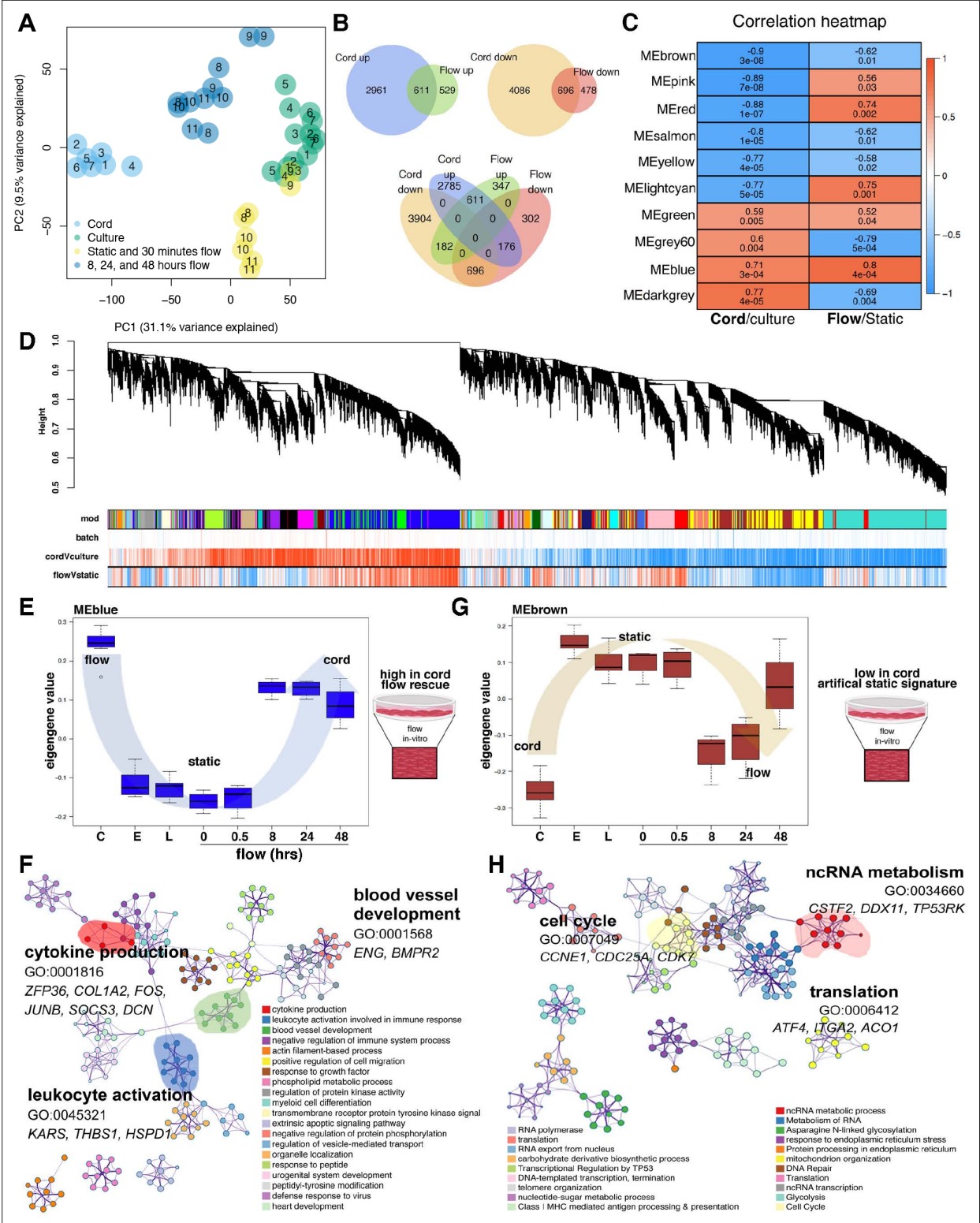

**Figure 3.** Flow rescues a degree of the cord transcriptome. (**A**) Principal component analysis (PCA) demonstrates stark differences in cord (in vivo) versus culture (in vitro) along PC1. Flow rescues the transcript from the culture toward the cord along PC1. Based on the transcriptional similarity of the different flow time points (**Figure 2B**) we consolidated the 'under flow' label for (**A**) for clarity. Based on the sample distribution on PC1 (from left to right: cord > extended flow > early culture + short-term flow > culture) and its magnitude (31% of the covariance in the dataset). PC1 primarily represents the differences between cord and culture samples (we interpret the PC2 to represent differences between short-term and extended flow). The middle

*Figure 3 continued on next page*

*Figure 3 continued*

position between cord and late culture is a partial rescue of the differences imparted by culture. (**B**) Venn diagram demonstrating the significant number of differentially expressed genes by condition and its concordant or discordant correlation to each another. (**C**) Correlation heatmap of top 10 module eigengenes (ME) by experimental condition, cord versus culture or flow versus static. The columns are labeled by experimental condition. The rows are labeled by the ME color. The biweight midcorrelation coefficients are shown numerically for each cell, with the significance of the correlation shown immediately below (false discovery rate, FDR). Cells are colored based on the strength and sign of the correlation. (**D**) Cluster dendrogram and module assignment for RNA modules from weighted gene co-expression network analysis (WGCNA). Identification of gene co-expression modules using average hierarchical linkage clustering. The vertical axis denotes the co-expression distance, and the horizontal axis corresponds to genes. Dynamic tree cutting was applied to identify modules by dividing the dendrogram at significant branch points. Modules are displayed with different colors in the horizontal bar immediately below the dendrogram, with gray representing unassigned genes. Correlation coefficients with experimental conditions are also represented based on strength and direction (negative correlations to positive correlations ranging from blue to red). (**E, F**) Eigengene value of flow-dependent rescue of the blue module; C = cord, E = early, L = late and enriched blue-module GEO. (**G, H**) Eigengene value of flow-dependent rescue of the brown module and enriched GEO.

The online version of this article includes the following figure supplement(s) for figure 3:

**Figure supplement 1.** Detailed Weighted Correlation Network Analysis (WGCNA) analysis by condition and sample.

profiled technical replicates of primary endothelial cells, primary SMCs (individual mono-cultures), and co-cultured endothelial cells and SMCs all plated to confluency using scRNAseq (***Supplementary file 6***, ***Figure 5—figure supplement 1***). Endothelial cells and SMCs were isolated from the same cord eliminating potential confounding factors associated with genetic variations. In total, 51,000 cells were sequenced with an average of 3402 genes and 18,740 transcripts per cell. Individual samples were independently analyzed to confirm correlation between triplicates, normalized and then combined for analysis. Unsupervised clustering demonstrated the cells cluster by origin (***Figure 5B–E***). We then confirmed cell clusters as endothelial cells (*PECAM1* and *CDH5*, ***Figure 5C, E***) and SMCs (*ACTA2* and *TAGLN*) (***Figure 5—figure supplement 1C***).

Transcriptomic profiles that defined each cluster was performed by Seurat and this information offered initial insight on transcriptional shifts that occurred consequently to heterotypic cell interactions. As shown by heatmap (***Figure 5F***), clear differences were noted when endothelial cells were in mono-culture (MC) versus co-culture (CC). Specifically, co-culture prompted a reduction in NOTCH target genes (*FABP4*, *GJA4*, *FABP5*, *HEY*) and a clear induction in TGFβ downstream targets (*SERPINE1*, *IGFBP7*, *SOX4*, *TIMP1*) (***Figure 5F***). Ingenuity pathway analysis provided further clarification as to the functional impact related to presence of SMCs. As shown in ***Figure 5G***, the major signaling pathways and transcriptional regulators that prompted transcriptional drifts on endothelial cells by co-culture included TGFβ, VEGF, TP53, HTT, MYC, TNF, EDN1, SP1, and HGF. We calculated a module score using the expression of downstream targets for TFGβ1 and VEGFA identified by ingenuity pathway analysis and found a significant increase upon co-culture for both (***Figure 5H***). This is entirely surprising as SMCs provide a source for these two cytokines. Activation of the TGFβ pathway results in shifts in extracellular matrix proteins, MMPs, and integrins (***Figure 5I***) and it is further supported by transcriptional increases in TGFβ receptors ACVRL1 and ENG. Interestingly, co-culture conditions resulted in an increase of clathrin-related genes (*AAK1*, *AP2B1*, and *CLTB*) and a decrease in caveolin-related genes (*CAV1* and *CAV2*) (***Figure 5J***). These changes occurred with no significant alterations in *CDH5*, *ERG*, *NOTCH1*, and *JAG1* (***Figure 5K***).

Naturally, the next question focused on which signatures impacted by heterotypic cell interactions yield a rescue of the in vivo condition. To delineate these transcriptional relationships, we overlapped scRNA sequencing data obtained from cord-derived endothelial cells and compared them to the mono- and co-culture endothelial transcriptomes (***Figure 6A–C***, ***Supplementary files 7 and 8***). Interestingly, global transcriptional profiling in uniform manifold approximation and projection (UMAP) showed a shift of co-culture toward cord (***Figure 6A***). In-depth analyses of the data using Seurat, GOs, and ingenuity pathways revealed cohorts of genes that were indeed rescued (either up- or downregulated) and genes that were not rescued by the co-culture condition. Examples of those categories are shown in ***Figure 6D*** and group analysis by dot blot as displayed in ***Figure 6E***. Genes rescued by co-culture relate to NOTCH signaling (*HES1*, *FABP4*) and TGFβ (*ENG*). In addition, we found that clathrin pathways, noted to be increased by SMC co-culture (***Figure 5***) were indeed part of the in vivo signature displayed by endothelial cells in the cord (***Figure 6F***) with upregulation of transcripts for *AAK1* and *EPN2*. Co-culture also was responsible for rescue of TJP1, responsible for tight junctions

**Table 2.** Most differentially up- and downregulated genes in cord (in vivo) and under flow (in vitro).

| Cord UP | | | Cord DOWN | | | Flow UP | | | Flow DOWN | | |
|---|---|---|---|---|---|---|---|---|---|---|---|
| Gene | t-stat | FDR | Gene | t-stat | FDR | Gene | t-stat | FDR | Gene | t-stat | FDR |
| SYNPO2 | 31.1 | 9E−14 | HIST1H3J | −30.7 | 9E−14 | MLKL | 19.8 | 2E−06 | GPX4 | −17.2 | 2E−06 |
| MMP28 | 28.5 | 2E−13 | RRM2 | −26.9 | 4E−13 | LPCAT4 | 16.3 | 3E−06 | ID3 | −15.6 | 3E−06 |
| ELN | 25.2 | 9E−13 | TK1 | −29.3 | 1E−12 | ZBTB11 | 15.2 | 3E−06 | AKT1 | −14.2 | 5E−06 |
| SORBS1 | 25.8 | 9E−13 | CCNA2 | −29.8 | 2E−12 | CTSL | 15.1 | 3E−06 | PDE4B | −13.2 | 8E−06 |
| SYTL2 | 31.6 | 9E−13 | HIST1H2BE | −22.9 | 2E−12 | SLC9A3R2 | 18.6 | 4E−06 | ORAI2 | −12.9 | 1E−05 |
| MYOCD | 24.5 | 1E−12 | NSD2 | −23.9 | 5E−12 | MEF2A | 13.7 | 6E−06 | MEX3D | −12.9 | 1E−05 |
| ID2 | 24.2 | 1E−12 | BUB3 | −21.3 | 6E−12 | VANGL1 | 14.2 | 8E−06 | GJA1 | −12.9 | 2E−05 |
| CRISPLD2 | 24.8 | 1E−12 | INCENP | −21.2 | 7E−12 | IGFBP5 | 14.1 | 1E−05 | COLEC12 | −11.8 | 2E−05 |
| TEK | 24.4 | 1E−12 | NCAPD2 | −27.2 | 7E−12 | PALM | 12.5 | 1E−05 | CCDC71 | −12.6 | 2E−05 |
| LMOD1 | 25.3 | 1E−12 | ASCC3 | −20.8 | 1E−11 | KCNN4 | 13.7 | 2E−05 | COPA | −10.3 | 5E−05 |
| WFDC1 | 28.7 | 1E−12 | BAX | −23.3 | 1E−11 | ORAI1 | 11.9 | 2E−05 | AMOTL2 | −9.7 | 7E−05 |
| NTRK3 | 23.4 | 2E−12 | HAGLR | −19.3 | 3E−11 | PPM1D | 12.3 | 2E−05 | PYCR3 | −10.4 | 7E−05 |
| CEBPD | 24.5 | 2E−12 | TOMM40 | −18.4 | 4E−11 | CAMK1 | 13.5 | 2E−05 | RBM12 | −9.7 | 7E−05 |
| MTURN | 24.4 | 3E−12 | RNPEP | −18.2 | 4E−11 | AL365205.1 | 14.3 | 2E−05 | MAP1S | −9.2 | 1E−04 |
| GHDC | 22.2 | 4E−12 | FJX1 | −20.9 | 5E−11 | CBLN2 | 11.7 | 2E−05 | FGFRL1 | −9.6 | 1E−04 |
| JUNB | 32.0 | 4E−12 | UBE2S | −20.7 | 5E−11 | NTHL1 | 11.4 | 2E−05 | LINC00704 | −10.9 | 1E−04 |
| ST6GALNAC5 | 22.3 | 4E−12 | SLC25A5 | −18.2 | 5E−11 | SLC2A3 | 11.5 | 2E−05 | PIM3 | −10.8 | 1E−04 |
| ID4 | 21.7 | 4E−12 | NCAPG2 | −17.8 | 5E−11 | TNXB | 11.2 | 2E−05 | PAWR | −10.0 | 1E−04 |
| CXCL2 | 21.5 | 5E−12 | LSS | −18.1 | 6E−11 | WNK1 | 11.6 | 3E−05 | ELAVL1 | −8.8 | 2E−04 |
| INMT | 21.3 | 5E−12 | KPNB1 | −18.1 | 7E−11 | CALCRL | 14.6 | 3E−05 | AAR2 | −8.7 | 2E−04 |
| FBLN2 | 21.1 | 6E−12 | PCCB | −18.7 | 8E−11 | RBPMS | 10.8 | 3E−05 | CENPJ | −9.2 | 2E−04 |
| ACTG2 | 30.0 | 7E−12 | RFWD3 | −17.1 | 8E−11 | SENCR | 13.5 | 3E−05 | GAB2 | −10.8 | 2E−04 |
| KLF4 | 20.6 | 8E−12 | CSE1L | −17.2 | 1E−10 | PSEN1 | 11.0 | 3E−05 | KRT10 | −9.3 | 2E−04 |
| LMCD1 | 21.0 | 8E−12 | APLN | −16.8 | 1E−10 | EPS15L1 | 12.3 | 3E−05 | YKT6 | −8.7 | 2E−04 |
| CTNNA3 | 20.8 | 9E−12 | MAP4K4 | −16.9 | 1E−10 | CFP | 11.8 | 3E−05 | MBD2 | −8.5 | 2E−04 |
| EMX2 | 26.7 | 9E−12 | FANCI | −17.4 | 2E−10 | CHN1 | 10.6 | 3E−05 | FAM168B | −12.0 | 2E−04 |
| ADGRA2 | 25.3 | 1E−11 | CENPN | −21.2 | 2E−10 | SH3BP2 | 10.3 | 5E−05 | NELFB | −9.6 | 2E−04 |
| SPEG | 20.0 | 1E−11 | PCNA | −18.4 | 2E−10 | SP1 | 10.3 | 5E−05 | TGFBRAP1 | −10.0 | 3E−04 |
| MMRN2 | 21.5 | 1E−11 | MDM2 | −18.8 | 2E−10 | EPHB4 | 11.5 | 5E−05 | APBA2 | −8.8 | 3E−04 |
| ACKR2 | 19.9 | 1E−11 | SSU72 | −17.9 | 2E−10 | CASKIN2 | 11.6 | 6E−05 | CHST7 | −10.0 | 3E−04 |

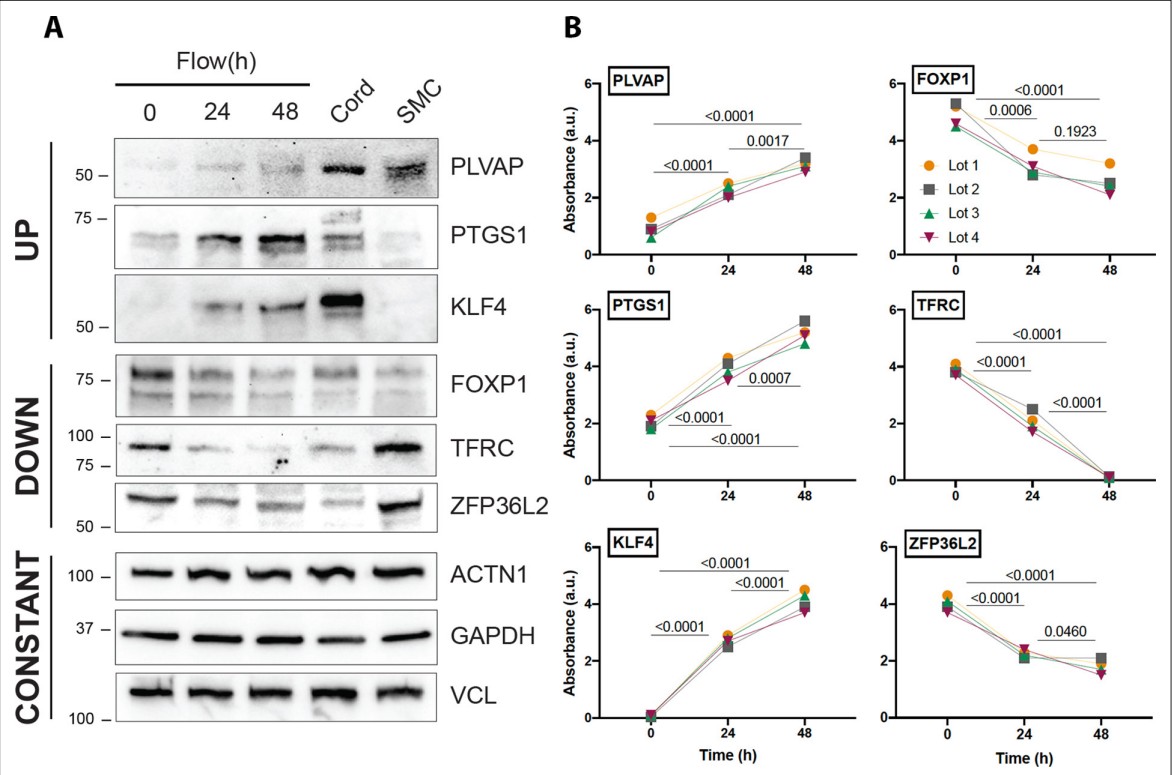

**Figure 4.** Protein validation of transcripts affected by flow. (**A**) Western blot analysis of three examples of transcripts that are regulated by flow in a concordant manner (upregulated, downregulated, and unchanged) in relation to cord isolated lysates. Uncropped data are shown in *Figure 4—source data 1*. (**B**) Quantification of the same genes using three independent biological replicates. Each color represents an independent experiment. Numbers show p value of analysis of variance (ANOVA) between the time points.

The online version of this article includes the following source data for figure 4:

**Source data 1.** Source file depicting uncropped western blot analysis from *Figure 4* (novel genes affected by flow).

and two transmembrane proteins that regulate calcium homeostasis (TMEM165 and 203) (*Figure 6F*). Interestingly we noticed a decrease in *IRF7* and *VASH1* under co-culture that also approximated the cord condition. In summary, co-culture of endothelial cells with SMCs normalized networks related to cell growth and differentiation, clathrin-vesicle-related genes, and recovered targets downstream TGFβ, recovering approximately 9% of the original cord (in vivo) signature (*Figure 7*).

## Discussion

Endothelial cells are characterized by a unique set of genes collectively referred to identity genes (i.e., *CDH5*, *PECAM1*, *ERG*) and a group of genes whose expression level varies according to stressors and environmental conditions. Precise information of both groups holds relevance to the interpretation of findings related to any experimental challenge. Despite the broad utilization of cultured endothelial cells, drifts in the transcriptional profiles upon expansion in vitro have not been rigorously addressed. Here, we undertook parallel transcriptomic analyses using genetically identical matches to determine the impact of the environment on cell culture and define whether specific signatures could be regained by changing environmental settings that will best approximate the native biological state.

To minimize confounding factors related to intrinsic genetic differences, we performed parallel transcriptomic profiling. Seven pairs of freshly isolated versus cultured endothelial cells were used for the initial profiles and the findings from these were validated against proteomics from seven independent pairs. Four additional cohorts were used to compare static versus flow versus freshly isolated conditions and single-cell RNAseq was subsequently used in the co-culture experiments. Our findings highlighted signatures that were uniquely associated with long-term exposure to shear stress in vitro that parallel expression profiles in vivo. We also identified signatures dependent on heterotypic,

**Table 3.** Concordant and discordant gene regulation in cord and flow conditions.

| Cord UP and Flow UP | | | | | Cord DOWN and Flow DOWN | | | | | Cord UP and Flow DOWN | | | | | Cord DOWN and Flow UP | | | | |
|---|---|---|---|---|---|---|---|---|---|---|---|---|---|---|---|---|---|---|---|
| | Cord | | Flow | | | Cord | | Flow | | | Cord | | Flow | | | Cord | | Flow | |
| Gene | t-stat | FDR | t-stat | FDR | Gene | t-stat | FDR | t-stat | FDR | Gene | t-stat | FDR | t-stat | FDR | Gene | t-stat | FDR | t-stat | FDR |
| ELN | 25.2 | 9E-13 | 8.3 | 2E-02 | TOMM40 | -18.4 | 4E-11 | -3.5 | 4E-02 | BMP6 | 22.0 | 8E-11 | -3.2 | 4E-02 | NECTIN1 | -15.5 | 3E-09 | 5.8 | 2E-03 |
| CRISPLD2 | 24.8 | 1E-12 | 5.7 | 2E-03 | FJX1 | -20.9 | 5E-11 | -5.9 | 2E-02 | IFT122 | 20.6 | 1E-10 | -9.8 | 2E-02 | BIN1 | -12.3 | 2E-08 | 8.3 | 3E-04 |
| TEK | 24.4 | 1E-12 | 9.1 | 7E-03 | MAP4K4 | -16.9 | 1E-10 | -4.2 | 3E-02 | CXCL3 | 16.1 | 2E-10 | -4.9 | 3E-02 | CAPRIN1 | -12.1 | 2E-08 | 4.1 | 2E-02 |
| FBLN2 | 21.1 | 6E-12 | 6.0 | 9E-03 | CENPN | -21.2 | 2E-10 | -4.4 | 1E-02 | ARHGEF9 | 16.0 | 4E-10 | -5.4 | 1E-02 | FBXO22 | -13.0 | 2E-08 | 3.7 | 4E-02 |
| KLF4 | 20.6 | 8E-12 | 10.9 | 1E-02 | KCTD5 | -18.1 | 3E-10 | -9.3 | 1E-03 | FBXO32 | 12.1 | 1E-08 | -9.3 | 1E-03 | EFNB1 | -13.1 | 2E-08 | 4.9 | 6E-03 |
| CTNNA3 | 20.8 | 9E-12 | 4.0 | 2E-02 | DKK1 | -19.5 | 5E-10 | -3.5 | 4E-02 | SIRPB2 | 13.6 | 5E-10 | -3.5 | 4E-02 | AKR1B1 | -14.0 | 3E-08 | 4.1 | 2E-02 |
| MMRN2 | 21.5 | 1E-11 | 8.9 | 4E-03 | FEN1 | -21.4 | 1E-09 | -4.4 | 2E-02 | CYP27A1 | 13.7 | 2E-08 | -4.6 | 2E-02 | GPSM2 | -12.2 | 3E-07 | 3.6 | 5E-02 |
| CMKLR1 | 21.4 | 1E-11 | 10.2 | 1E-02 | DTYMK | -20.9 | 1E-09 | -3.8 | 3E-02 | GUCY1A2 | 11.7 | 2E-08 | -4.2 | 3E-02 | INTS13 | -12.8 | 8E-07 | 3.7 | 3E-02 |
| DLL1 | 20.2 | 2E-11 | 5.9 | 3E-02 | SEC61B | -15.3 | 2E-09 | -5.7 | 2E-03 | MLLT6 | 11.0 | 3E-08 | -5.3 | 2E-03 | EYA3 | -10.8 | 9E-07 | 5.6 | 5E-03 |
| IGFBP5 | 23.1 | 2E-11 | 14.1 | 1E-05 | RANGAP1 | -15.7 | 3E-09 | -4.3 | 1E-02 | ZNF365 | 10.4 | 7E-08 | -4.6 | 1E-02 | HMGCR | -9.0 | 2E-06 | 3.7 | 3E-02 |
| SRL | 19.4 | 3E-11 | 6.1 | 3E-02 | DDX52 | -14.5 | 4E-09 | -4.0 | 2E-02 | HMCN2 | 15.9 | 1E-07 | -4.7 | 2E-02 | ARFGEF2 | -9.5 | 3E-06 | 4.1 | 2E-02 |
| FGF18 | 22.1 | 3E-11 | 8.9 | 3E-03 | LINC01013 | -24.7 | 5E-09 | -3.9 | 3E-02 | DACT3 | 11.0 | 1E-07 | -4.6 | 3E-02 | DHX9 | -9.1 | 4E-06 | 3.3 | 4E-02 |
| AQP1 | 20.9 | 4E-11 | 6.0 | 2E-02 | RAB1B | -13.1 | 8E-09 | -4.4 | 2E-02 | PAM | 9.6 | 3E-07 | -3.7 | 2E-02 | LYAR | -9.8 | 4E-06 | 3.8 | 3E-02 |
| STOM | 18.0 | 4E-11 | 9.0 | 6E-04 | ATAD2 | -12.2 | 8E-09 | -3.2 | 5E-02 | LIMCH1 | 10.0 | 3E-07 | -4.9 | 5E-02 | TRIM7 | -7.6 | 4E-06 | 6.7 | 9E-03 |
| TNXB | 18.4 | 5E-11 | 11.2 | 2E-05 | CDCA4 | -14.5 | 9E-09 | -4.6 | 1E-02 | ADSSL1 | 9.8 | 9E-07 | -4.6 | 1E-02 | NDUFB10 | -10.5 | 5E-06 | 4.2 | 1E-02 |
| SMOC2 | 21.6 | 5E-11 | 4.1 | 3E-02 | CENPO | -23.0 | 9E-09 | -5.6 | 4E-03 | SYNGR2 | 8.4 | 1E-06 | -5.6 | 4E-03 | HIST2H2BF | -9.7 | 8E-06 | 3.8 | 3E-02 |
| JPH4 | 16.8 | 1E-10 | 15.2 | 1E-03 | AP1B1 | -14.8 | 1E-08 | -8.3 | 9E-04 | CCL2 | 8.7 | 1E-06 | -8.3 | 9E-04 | ZNF185 | -10.3 | 8E-06 | 4.1 | 2E-02 |
| NTN1 | 18.3 | 1E-10 | 6.1 | 2E-03 | PGF | -12.5 | 1E-08 | -5.0 | 3E-02 | PDE1C | 8.5 | 1E-06 | -5.0 | 3E-02 | NQO1 | -15.2 | 9E-06 | 10.6 | 1E-04 |
| ANGPTL1 | 18.2 | 1E-10 | 6.5 | 1E-03 | ADAM9 | -14.1 | 1E-08 | -5.3 | 5E-03 | SELENOP | 8.6 | 1E-06 | -5.3 | 5E-03 | PLCB3 | -10.6 | 9E-06 | 3.4 | 3E-02 |
| PTPN13 | 16.7 | 2E-10 | 4.0 | 2E-02 | CHAF1A | -17.8 | 1E-08 | -3.7 | 2E-02 | RHOB | 10.8 | 2E-06 | -3.7 | 2E-02 | ELMOD1 | -6.9 | 1E-05 | 3.6 | 3E-02 |
| PLPP3 | 16.5 | 2E-10 | 10.0 | 4E-03 | DLAT | -13.3 | 2E-08 | -3.6 | 3E-02 | SORBS2 | 8.0 | 2E-06 | -4.0 | 4E-02 | SAAL1 | -10.7 | 1E-05 | 4.6 | 8E-03 |
| ST8SIA6 | 17.0 | 2E-10 | 12.7 | 4E-03 | C19orf48 | -19.9 | 2E-08 | -4.1 | 3E-02 | FAM198B | 9.1 | 4E-06 | -4.5 | 2E-02 | SLC25A19 | -8.7 | 2E-05 | 3.9 | 2E-02 |
| MMP24 | 17.4 | 2E-10 | 4.4 | 1E-02 | TUBB | -11.5 | 2E-08 | -3.3 | 4E-02 | CD83 | 8.5 | 4E-06 | -3.6 | 3E-02 | CDC25B | -8.5 | 2E-05 | 10.2 | 4E-03 |
| PLCB4 | 17.4 | 2E-10 | 5.6 | 3E-02 | H2AFZ | -15.6 | 2E-08 | -3.2 | 4E-02 | TMEM184A | 9.4 | 4E-06 | -3.8 | 2E-02 | ZC3H14 | -7.0 | 2E-05 | 4.5 | 3E-02 |
| PARM1 | 15.5 | 3E-10 | 6.2 | 3E-03 | MEX3A | -12.1 | 2E-08 | -5.1 | 7E-03 | ARHGEF37 | 8.7 | 5E-06 | -4.6 | 9E-03 | SCD | -7.9 | 2E-05 | 4.0 | 4E-02 |
| RAMP3 | 15.2 | 5E-10 | 3.1 | 5E-02 | SPDL1 | -16.1 | 2E-08 | -4.1 | 2E-02 | EGR1 | 12.1 | 6E-06 | -5.5 | 3E-03 | MIR100HG | -7.2 | 2E-05 | 6.8 | 8E-04 |
| HIPK3 | 20.9 | 5E-10 | 4.7 | 1E-02 | NME4 | -19.3 | 3E-08 | -4.6 | 2E-02 | ARSG | 11.1 | 6E-06 | -4.9 | 9E-03 | SCARB1 | -11.9 | 2E-05 | 4.2 | 3E-02 |
| NOV | 15.1 | 5E-10 | 8.9 | 7E-04 | USP31 | -11.8 | 3E-08 | -4.0 | 5E-02 | MEFV | 9.7 | 7E-06 | -3.4 | 5E-02 | MIR34AHG | -9.5 | 2E-05 | 4.8 | 6E-03 |
| GLIS3 | 16.8 | 6E-10 | 4.4 | 2E-02 | DNAH11 | -15.5 | 3E-08 | -3.8 | 3E-02 | NUAK1 | 11.4 | 8E-06 | -3.8 | 4E-02 | CNRIP1 | -9.0 | 3E-05 | 3.9 | 2E-02 |
| ITPR1 | 27.3 | 7E-10 | 5.9 | 3E-02 | UBE2N | -11.1 | 3E-08 | -5.6 | 3E-02 | CSRNP1 | 17.7 | 9E-06 | -4.7 | 2E-02 | CCDC51 | -8.7 | 3E-05 | 3.9 | 3E-02 |

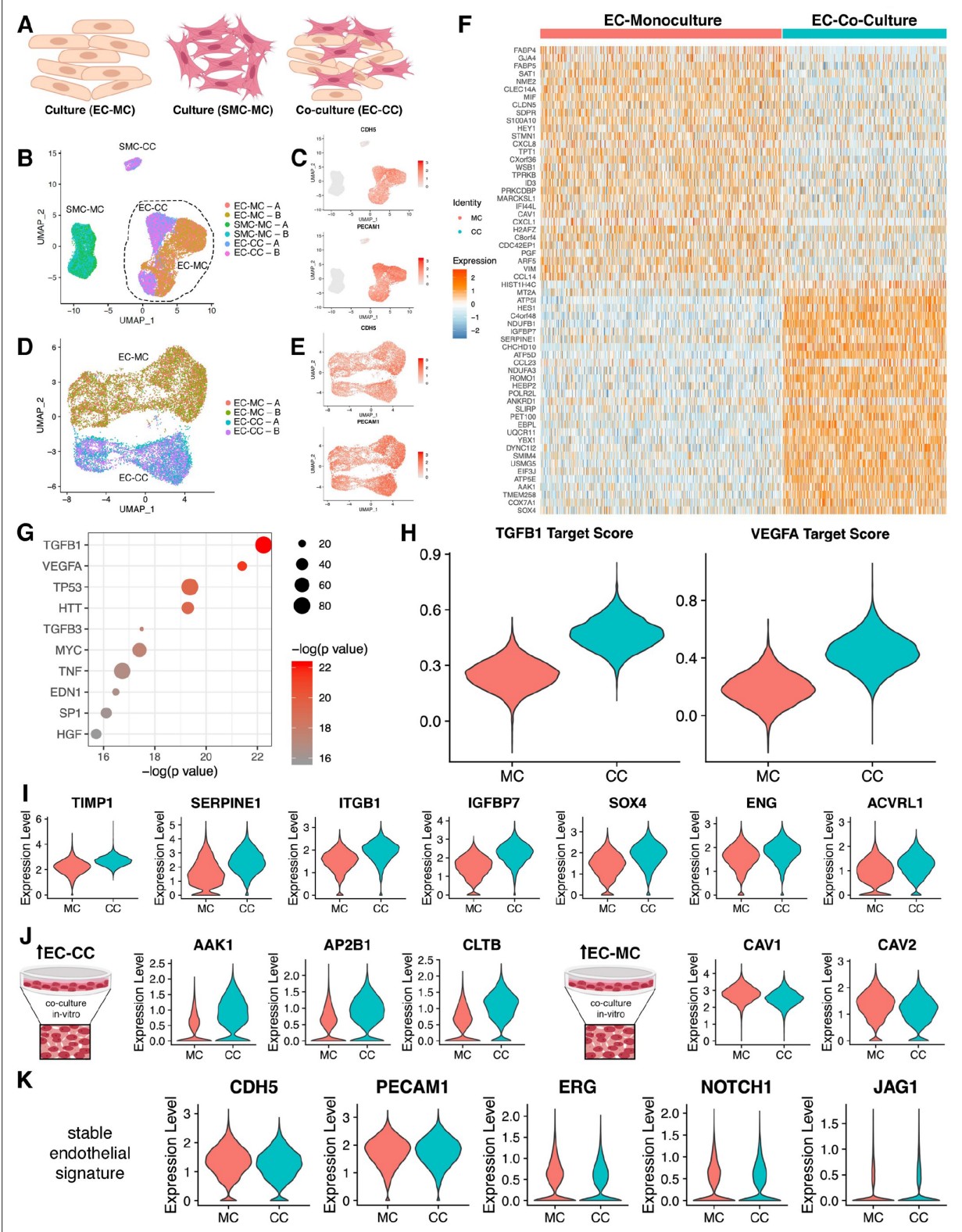

**Figure 5.** Endothelial cell–smooth muscle cell interactions. (**A**) Schematic overview of single-cell RNA sequencing (RNAseq) experiments. (**B**) Uniform manifold approximation and projection (UMAP) of scRNAseq data with four distinct clusters with two technical replicates (labeled A/B) as indicated in the legend. (**C**) Identity of endothelial cells was confirmed by expression of CDH5 and PECAM transcripts. (**D**) UMAP of scRNAseq for endothelial cell mono-culture (EC-MC) and endothelial cell co-culture (EC-CC) with biological replicates as indicated. (**E**) Identity of endothelial cells was confirmed

*Figure 5 continued on next page*

*Figure 5 continued*

by expression of CDH5 and PECAM transcripts. (**F**) Heatmap identifying the top differentially expressed genes with log fold >2 for each condition relative to the other cell types. (**G**) Ingenuity analysis demonstrates most significantly upregulated module score based on growth factors, cytokines, and transcription factors. (**G**) TGFB1 and VEGF show the highest module score in co-culture relative to endothelial cell monoculture. (**I**) TGFB1 activated genes are upregulated in co-culture. (**J**) Clathrin family members are upregulated in co-culture; whereas caveolin family members are decreased in co-culture. (**K**) Endothelial cell makers are unchanged and stable in mono- and co-culture endothelial cells.

The online version of this article includes the following figure supplement(s) for figure 5:

**Figure supplement 1.** Single-cell RNAseq: endothelial cell–smooth muscle cell interactions.

endothelial cell–SMC interactions that were lost in vitro, but a hallmark of the in vivo state. The findings offer an important resource to query how expression profiles of specific genes change in relation to a subset of environmental conditions.

A major adaptation that cells must acquire when placed in culture relates to cell proliferation. Once seeded, endothelial cells undergo significant expansion that is thought to be attenuated or suppressed at confluency. Nonetheless, we demonstrate that high levels of transcripts related to cell cycle, mitosis, and DNA repair mechanisms are still present at confluency and represent the single most significant alteration when comparing freshly isolated cells to genetically identical cohorts in vitro. Similarly, there are significant alterations in cytoskeletal dynamics and focal adhesions that are artificially elevated in vitro, compared to ex vivo.

Recapitulating the native flow seen by endothelial cells by exposure of static cultures to shear stress resulted in a significant shift toward ex vivo (freshly isolated cells) signature. Much has been done to understand transcriptional responses to flow. Most of these have been focused on early responses in the absence of in vivo genetically matched counterparts (*Ajami et al., 2017*; *Chen et al., 2001*; *Chu and Peters, 2008*; *Conway et al., 2010*; *Dekker et al., 2002*; *Guo et al., 2007*). Previous studies have described the effects of cell culture on endothelial cells with changes in gene expression (*Burridge and Friedman, 2010*; *Lacorre et al., 2004*; *Sabbagh and Nathans, 2020*; *Shima et al., 1995*) or characterized the change in differential gene expression with shear stress (*Brooks et al., 2004*; *Frueh et al., 2013*; *Maurya et al., 2021*). The novelty of our study is the systematic analysis of 'recovery' and 'not recovery' based on changes in shear stress and exposure to SMCs.

Our data found agreement with previous findings of short time exposure to shear stress we noted an impressive induction of KLF2 and KLF4 (*Coon et al., 2022*). However, longer exposure to laminar flow (8, 24, and 48 hr) progressively increased the resemblance to the in vivo transcriptome, as noted by correlation coefficients. Specifically, we found that two major pathways and their downstream genes were regained by long-term flow: BMP and NOTCH signaling. Importantly, it has been recently shown that BMP signaling is significantly potentiated by flow (*Baeyens et al., 2016*). Indeed, several SMAD targets were rescued by incorporating long-term flow into cultures. Similarly, NOTCH target genes (*HES*, *HEY*) regained levels similar to those captured in freshly isolated preparations. These findings are congruent with recent studies demonstrating that NOTCH signaling was increased by flow and mechanosensing (*Mack et al., 2017*). An unexpected GO signature regained by shear stress included proteins associated with cellular localization, such as ITPR1, IGFBP5, DLL1, among others, highlighting the role of laminar shear stress in endothelial cell polarity. Not surprisingly, the most significant protein changes coincide with significant corresponding changes in RNA but the most significant changes in RNA did not coincide with significant changes in corresponding protein levels (*Supplementary file 9*).

Alterations in junctional proteins and cytoskeletal architecture were recovered in endothelial cell–SMC co-cultures. Co-culture of endothelial cells with SMCs also induced TGFβ downstream targets in the endothelium, including several extracellular matrix proteins and integrins which brought further alignment to the in vivo transcriptome. In addition, SMCs significantly reduced the prominent proliferative signature of endothelial cells and promoted a partial recovery in endothelial cell differentiation. Specifically, this included ENG and integrins regulated by TGFB1 (ITGB1, ITGA1, ITGA5), as well as several extracellular membrane proteins (COL1A1, FN1, TIMP1, SERPINE1) (*Gallicchio et al., 1994*; *Nackman et al., 1996*; *Powell et al., 1998*). It can be postulated that loss of architecture in vitro could induce the loss of expression of acute phase transcripts, as seen with injury of the aorta in vivo (*Shirali et al., 2018*). These endothelial-heterotypic crosstalk have been shown essential during development and altered in vascular pathologies such as aneurysms (*Boezio et al., 2020*).

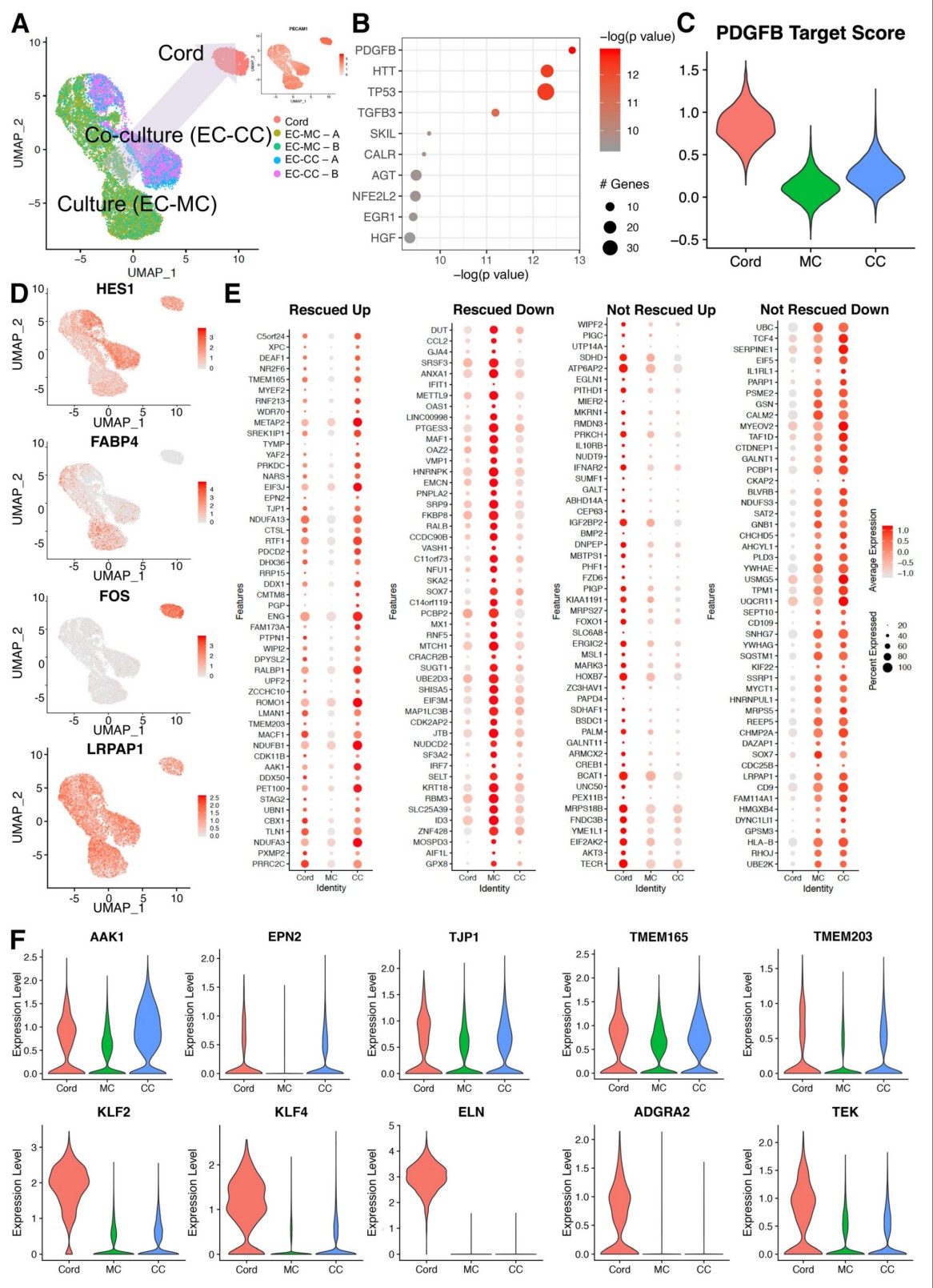

**Figure 6.** Co-cultured endothelial cells with smooth muscle cells rescue a cohort of genes when compared to the cord transcriptome. (**A**) Uniform manifold approximation and projection (UMAP) of endothelial cell co-culture (EC-CC) with smooth muscle cells versus endothelial cell monoculture (EC-MC) in relation to endothelial cells isolated directly from umbilical cord. Insert: confirmed endothelial cell identity by PECAM. (**B**) Ingenuity analysis demonstrates most significantly upregulated module score based on growth factors, cytokines, and transcription factors. (**C**) PDGFB, the most

*Figure 6 continued*

significantly upregulated growth factor, is rescued by co-culture. (**D**) Environment-dependent transcriptional enrichment demonstrated by UMAP. (**E**) Dotplot illustrates the top markers of in cord, monoculture (MC), and co-culture (CC). Dot size corresponds to the proportion of cells within the group expressing each transcript and dot color intensity corresponds to the expression level. (**F**) Violin plot of environment-dependent (heterotypic co-culture) gene expression illustrating examples of genes rescued (AKK1, EPN2, TJP1, TMEM16S, TMEM203) and non-rescued genes (KLF2, KLF4, ELN, ADGRA2, TEK).

Exposing endothelial cells to culture conditions does not appear to significantly affect cellular identity. Transcriptional levels of CDH5, PECAM1, ERG, Claudins, Sox(s), and other so-called endothelial markers were not significantly impacted. ERG is essential for regulation of CDH5, VWF, and NOS3 as well as a hallmark of endothelial cell lineage (*Birdsey et al., 2008*; *Laumonnier et al., 2000*; *Nikolova-Krstevski et al., 2009*; *Shah et al., 2016*; *Yuan et al., 2009*).

Despite notable strengths, our study has several limitations, including the individual participant heterogeneity which introduces inter-subject variability and lack of functional read-outs of the biology described. We acknowledge this limitation, albeit patient diversity provides a more representative and realistic model to understand biology and disease. It should be also stressed that the present study focuses on HUVECs and does not delve into the remarkable heterogeneity of organ-specific vascular beds that might respond differently to shear stress. Additionally, we and others *Kalluri et al., 2019* have noticed distinct populations in the PCA of cultured ECs during our single-cell RNAseq studies that were not explored here. Evaluation as to these subpopulations, which are also noted in the aorta in vivo (*Kalluri et al., 2019*) reflect transcriptionally distinct groups or different states of cyclic expression patterns and requires a more thorough analysis and lineage tracing studies that are distinct from the objective of the question posed here. This second type of analysis, along with assessment of chromatin states (ATACseq) may provide clear-cut cell-subtype and state-specific information. Finally, considerations of how in vivo metabolites influence the transcriptional read-out of the endothelium were not explored here. It is likely that metabolites may aid in further correcting shifts from in vivo to in vitro conditions that were not affected by the two factors evaluated here. We found that 26% out the 43% of transcriptional alterations could not be recovered by either shear stress (which rescued 17% of the changes) or by contact with SMCs (that rescued 9% of the changes). There are still 17% of transcriptional drifts that could not be recovered.

The ability to grow and study endothelial cells in vitro has enabled investigators to ask questions under well controlled, yet artificial, conditions. The consequences associated with phenotypic alterations of ex vivo expanded cells remain unknown despite ample evidence that culture conditions

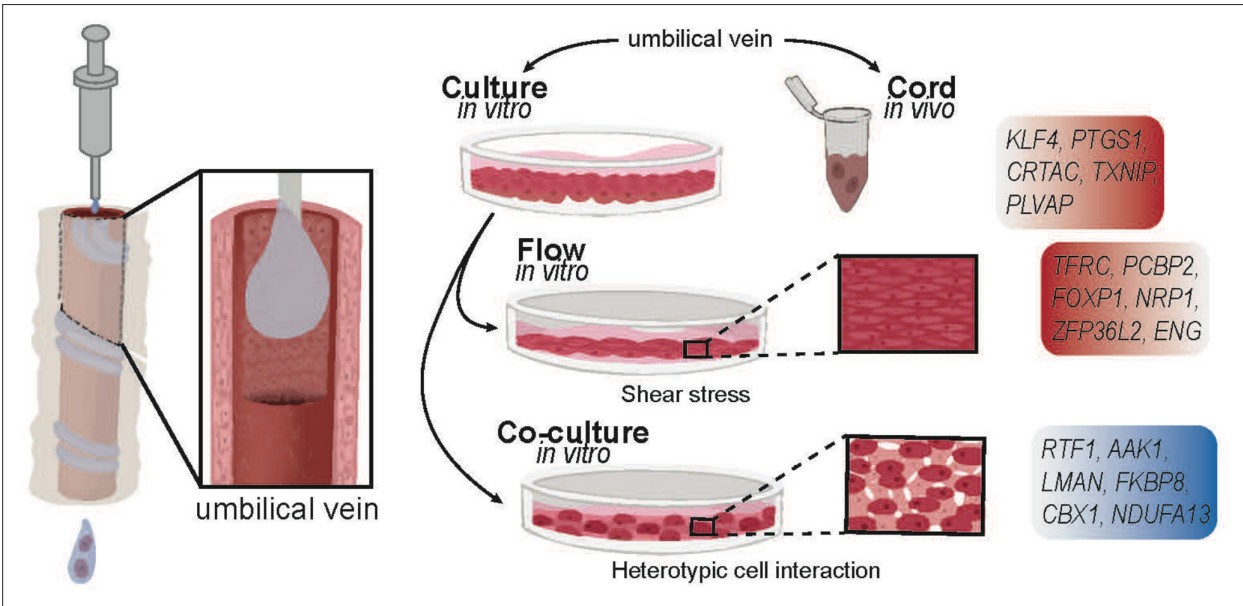

**Figure 7.** Summary figure. Schematic representing experimental design, culture conditions, and corresponding validated genes changes.

exert profound influence upon cellular biological properties (*Boquest et al., 2005*; *Bork et al., 2010*; *Forsyth et al., 2008*; *Roobrouck et al., 2011*). We defined a transcriptionally unique fingerprint of endothelial cells immediately removed from the cord and mapped how environmental changes uniquely impact this profile. These -omics analyses offer information that can guide researchers to have a better understanding of intrinsic mechanisms that are not captured when studying signaling pathways and molecular processes in culture. Appreciating these nuances and recapitulating intrinsic shear stress and heterotypic cell interactions will help propel reprogramming efforts for the generation of a more representative in vivo model system allowing investigators to better interpret genetic modifiers that affect or are affected by endothelial cells.

## Methods

### Endothelial cell isolation, culture, and RNA extraction

Human umbilical cords were collected under Institutional Review Board (UCLA IRB#16-001694) at time of birth. Umbilical cords were processed within 2–4 hr from time of birth and cells were isolated as previously described (*Crampton et al., 2007*). All samples were collected from participants who provided signed written informed consent and were de-identified immediately after cords were obtained. *Table 1* describes the clinical details of the participants/donors. The umbilical cord was clamped bilaterally, cut at least into two fragments, and placed in HBSS at room temperature. The umbilical vein was cannulated with a 18 G animal feeding needle with a blunt tip in the direction of oxygenated blood flow from the placenta to the fetus. Subsequently, the umbilical vein was serially washed with 20 ml of Hanks' Balanced Salt Solution (HBSS) (HBBS) 3× to remove blood cells from the lumen. For collection of in vivo samples: 1 ml of RLT from RNeasy Micro Kit (QIAGEN, Germantown, MD) flushed through closed circuit and re-aspirated with the distal end of the umbilical cord clamped and stored in −80°C until all RNA was ready to be extraction. The length of time to obtained cells was approximately 30–60 min from cord clamping at delivery/birth.

For isolation of endothelial cells for culture, the other half of the umbilical cord was flushed with 8 ml of collagenase-2 (210 IU, Worthington Biochemical, Lakewood, NJ) and the exudate was further incubated at 37°C for 20 min to dissociate cells. Collagenase was inactivated with the addition of equivalent volume 10% fetal bovine serum (FBS) with MCDB 131 media (VEC Technologies, Rensselaer, NY). The cells were pelleted, resuspended in media, plated and cultured for 30 min on tissue culture treated dishes in humidified incubator at 37°C and 5% $CO_2$. After the short incubation period (30 min), the plates were washed to remove non-adherent cells. After cells became confluent, additional purification steps were conducted (Miltenyi Biotec #130-091-935) and the purity of the endothelial isolates was evaluated by FACs analysis, immunocytochemistry and, in the case of the in vivo counter parts (RNA from the umbilical cords) we also used scRNAseq. The information on the purity of the isolated cells is shown in *Figure 1—figure supplement 1*. For culture cells, they were passaged using 1× Trypsin (Fisher Scientific, Waltham, MA) and collected with RLT for RNA extraction at early passage (passage 2–3) and at late passage (passage 7–8). Subsequently all RNA was extracted in tandem using RNeasy Micro Kit (QIAGEN, Germantown, MD). Contamination with genomic DNA was eliminated with incubation of DNase I at room temperature. Agilent Bioanalyzer 2100 system (Agilent Technologies, Santa Clara, CA) was used to assess RNA integrity and Qubit (Invitrogen, Carlsbad, CA) for RNA concentration and purity.

For co-culture experiments, primary umbilical SMCs were isolated from the same genetically identical cords after removal of the endothelium and expanded in vitro for two passages. The purity of the SMCs was tested by scRNAseq (*Figure 1—figure supplement 1*). SMCs were then seeded at 50% confluency onto the culture plate 24 hr prior to adding HUVECs to reach a confluent density. After an additional 24 hr co-cultured cells were trypsinized for scRNAseq experiments.

### Sequencing data samples and mapping

Library preparation was performed using TruSeq Total RNASeq Kit (Illumina, San Diego, CA) according to the manufacturer's instruction. Sequencing was conducted on an Illumina HiSeq 4000 (RNAseq) and NovaSeq S2 (scRNAseq) instrument (Illumina, San Diego, CA) at the University California Los Angeles. Sequencing parameters were optimized for 50 bp single-end reads at a depth of 30,000 million reads/ sample. Reads were mapped to the hg38 build of the human genome with Bowtie2 (*Langmead and*

*Salzberg, 2012*) and RNAseq reads were mapped with STAR (*Dobin et al., 2013*). RNAseq experiments that measured accessibility and expression in different environment (cord versus culture) were all conducted at least twice. Benjamini–Hochberg false discovery rate (FDR) method was used to correct for all multiple testing in this study with a significance threshold of FDR <0.05. No explicit power analysis was used to compute sample size.

## RNAseq gene expression analysis

Gene expression analysis was conducted using R software. First, each raw count TPM gene expression profile was log10 transformed and rescaled to zero-mean and unit-resolution for both the cord versus culture and flow versus static datasets. Data were adjusted for batch effects using an empirical Bayes framework with the ComBat function from the sva package; no covariates were included in the model and the algorithm was set to use non-parametric adjustments. The expression of individuals genes was screened for associations with experimental treatments using biweight midcorrelation, a robust correlation measure, with the bicorAndPvalue function from the WGCNA package. Individual genes were also tested for associations with experimental treatments using Welch's *t*-test using the base R *t*-test function, adopting a Bonferroni-corrected significance threshold (p < 2.5e−6). Principal components analysis (PCA) was conducted following gene-wise scaling to zero-mean and unit-variance. WGCNA was conducted using the blockwiseModulesvfunction from the WGCNA package; the network soft-thresholding power was set to 3, the network type was set to 'signed hybrid'; and the entire gene set was used for module detection by adjusting the maxBlockSize. The data can be found on the Gene Expression Omnibus (GEO) under the GEO accession number GSE158081. Both STRINGv10 (*Szklarczyk et al., 2015*) and Metascape (*Zhou et al., 2019*) were used to generate differential gene expression figures.

## Flow Profiler: transcriptomic web application

The web application for viewing flow transciptomic data was built to display data as a table or plot and allows for easy accessibility to all investigators. The application was developed using html, JavaScript, and CSS. A plot is drawn for each gene, at the indicated times and the average value of all available samples is displayed. A searchable/filtrable table gather the average values per gene plus the origin and slope of the resulting curve. Origin and slope were computed using the linear regression functions from Excel 2019 (INTERCEPT and SLOPE). The website provides a set of functionalities supporting the analysis of gene profiles. In addition to basic tools allowing search, filter, and combination of multiple profiles on one plot, two tools allow to find the most similar or divergent profiles compared to a selection of one or more profiles. The similarity is defined by a value between 0 and 2. Then a range is defined for each available time point (the average value of the selected gene plus and minus the similarity value). All profiles with all of their timespoints within this range are filtered. Divergence works in a comparable way, but only profiles with an average value outside the defined range are filtered. The two functions can also be used with an artificial curve as the comparison basis (the user can draw the desired curve on the plot).

## LC–MS-based proteomics

Protein samples were reduced and alkylated using 5 mM Tris (2-carboxyethyl) phosphine and 10 mM iodoacetamide, respectively, and digested by the sequential addition of trypsin and lys-C proteases, as described (*Wohlschlegel, 2009*; *Florens et al., 2006*). The digested peptides were desalted using Pierce C18 tips (Thermo Fisher Scientific, Waltham, MA), dried and resuspended in 5% formic acid, and fractionated online using a 25 cm long, 75 µM inner diameter fused silica capillary packed in-house with bulk C18 reversed phase resin (1.9 µM, 100 A pores, Dr. Maisch GmbH). The 140-min water–acetonitrile gradient was delivered using a Dionex Ultimate 3000 UHPLC system (Thermo Fisher Scientific, Waltham, MA) at a flow rate of 300 nl/min (Buffer A: water with 3% Dimethyl sulfoxide (DMSO) and 0.1% formic acid and Buffer B: acetonitrile with 3% DMSO and 0.1% formic acid). Peptides were ionized by the application of a distal 2.2 kV and introduced into the Orbitrap Fusion Lumos mass spectrometer (Thermo Fisher Scientific, Waltham, MA) and analyzed by tandem mass spectrometry (MS/MS). Data were acquired using a Data-Dependent Acquisition (DDA) method comprised of a full MS1 scan (Resolution = 120,000) followed by sequential MS2 scans (Resolution = 15,000) to utilize the remainder of the 3-s cycle time. The mass spectrometry proteomics data have

been deposited to the ProteomeXchange Consortium via the PRIDE (*Perez-Riverol et al., 2019*) partner repository with the dataset identifier PXD020958 and 10.6019/PXD020958. Data analysis was performed using the MSGF+ search engine (*Kim and Pevzner, 2014*) via the target-decoy strategy against the EMBL Human reference proteome (UP000005640 9606). The identification false detection rates (FDRs) at the peptide-spectrum-match (PSM) were defined using Percolator, protein identification confidence was estimated via the stand-alone implementation of FIDO such that analytes had respective *q*-values at or below 0.01 at both PSM and protein level (*Serang et al., 2010*; *Granholm et al., 2014*; *The et al., 2016*). Extracted ion chromatograms were calculated for each peptide using Skyline (*MacLean et al., 2010*). The MSStats R-package was used to normalize across runs using quantile normalization, summarize peptide-level intensities into a protein-level abundance, and perform statistical testing to compare protein abundance across conditions (*Choi et al., 2014*).

## Shear stress application

Confluent endothelial monolayers were grown on tissue culture treated 6-well plates (Falcon #08-772-1B) in complete MCDB-131 media (VEC Technologies # MCDB131-WOFBS) plus 10% FBS (Omega Scientific #FB-11) containing 4% dextran (Sigma-Aldrich #31392) for approximately 12–18 hr and then subjected to shear stress (130 rpm) in new medium containing 4% dextran (Sigma-Aldrich #31392) for indicated time intervals and cultured alongside static controls. Orbital shear stress (130 rpm) was applied to confluent cell cultures by using an orbital shaker positioned inside the incubator as previously discussed (*Dardik et al., 2005*). The shear stress within the cell culture well corresponds to arterial magnitudes (11.5 dynes/cm$^2$) of shear stress. To reduce issues associated with uniformity of shear stress, the endothelial cell monolayers in 6-well plates were lysed after removing center region using cell scraper (BD Falcon #35-3085) and washing with 1× HBSS (Corning #21-022-CV). A 1.8 cm blade was used circumferentially to remove the center of the monolayer that did not see the higher shear stress.

## Single-cell sequencing and data analysis

Single cells were isolated from umbilical cord flushes as described above. To keep the processing time between tissue harvesting and single-cell lysis at a minimum, no further cell type enrichment step was performed. For the generation of single-cell gel beads in emulsion, cells were loaded on a Chromium single-cell instrument (10× Genomics, Pleasanton, CA) with an estimated targeted cell recovery of ~5000 cells as per manufacturer's protocol. In brief, single-cell suspension of cells in 0.4% bovine serum albumin–phosphate-buffered saline were added to each channel on the 10× chip. Cells were partitioned with Gel Beads into emulsion in the Chromium instrument where cell lysis and barcoded reverse transcription of RNA occurred following amplification. Single-cell RNAseq libraries were prepared by using the Chromium single cell 3′ library and gel bead kit v3 (10× Genomics, Pleasanton, CA). Sequencing was performed (as described above) and the digital expression matrix was generated by demultiplexing, barcode processing, and gene unique molecular index counting by using the Cell Ranger pipeline (10× Genomics, Pleasanton, CA). The data can be found under the GEO accession number: GSE156939.

To identify different cell types and find signature genes for each cell type, the R package Seurat (version 3.1.2) was used to analyze the digital expression matrix. Cells with less than 500 unique molecular identifiers (UMIs) and greater than 50% mitochondrial expression were removed from further analysis. Seurat function NormalizeData was used to normalize the raw counts. Variable genes were identified using the FindVariableGenes function; genes with normalized expression values between 0.1 and 5 and with a dispersion of at least 0.5 were considered variable. The Seurat ScaleData function was used to scale and center expression values in the dataset for dimensional reduction. PCA, *t*-distributed stochastic neighbor embedding (t-SNE), and UMAP were used to reduce the dimensions of the data, and the first two dimensions were used in plots. A graph-based clustering approach was later used to cluster the cells; then signature genes were found and used to define cell types for each cluster. ECs were selected based on high expression of *PECAM1* and *CDH5* genes. SMCs were identified by the high expression of *ACTA2* and *TAGLN* genes. Module scores were calculated using the AddModuleScore function with default parameters.

## Western blots

Endothelial cells were lysed in modified Radioimmunoprecipitation assay (RIPA) buffer (50 mM Tris pH 8, 0.1% sodium dodecyl sulfate (SDS), 0.5% sodium desoxycholate, 1% Triton X-100, 150 mM NaCl, 1× protease inhibitor cocktail). Proteins were separated by SDS–polyacrylamide gel electrophoresis gradient (4–20%) gel and transferred onto nitrocellulose membranes and incubated overnight at 4°C with primary antibodies. The following primary antibodies were used in this study: PLVAP (DSHB, Cat#MECA-32); PTGS1 (Cell Signaling Technologies, Cat#9896S); TXNIP (Cell Signaling, Cat#71632); FOXP1 (Cell Signaling Technologies Cat#4402S); TFRC (DSHB, Cat#G1/221/12); ZFP36L2 (Cell Signaling, Cat#2119); ACTN1 (Sigma, Cat#A5044); GAPDH (Millipore Sigma, Cat#MAB374); VCL (Millipore Sigma, Cat#V-9131). Secondary antibodies included: Amersham ECL Rabbit IgG HRP-Linked Whole Antibody (Cat#NA934) and Amersham ECL Mouse IgG, HRP-Linked Whole Antibody (Cat#NA931) both from Cytiva. Immuno-complexes were detected by enhanced chemiluminescence with SuperSignal West Pico PLUS (Thermo Fisher Scientific #PI34580) and Femto Maximum Sensitivity Substrate (Thermo Fisher Scientific #PI34096) using ChemiDoc Touch Imaging System (Bio-Rad Laboratories). Quantification of bands by densitometry analysis was performed using ImageLab Software (Bio-Rad Laboratories).

## Acknowledgements

This work was supported by a grant from the National Institutes of Health R35HL140014 (MLIA), Leducq Foundation (MLIA and MV), R01HL147187 (CER), FAPESP 2016/19968-3 (VF), and K12 HD000849 (YA), awarded to the Reproductive Scientist Development Program by the Eunice Kennedy Shriver National Institute of Child Health & Human Development, by the American College of Obstetricians and Gynecologists, as part of the Reproductive Scientist Development Program (YA), and the Ruth L Kirschstein National Research Service Award T32HL069766 (YA). We thank Bill Brancart for his contributions to the Flow Profiler.

## Additional information

### Funding

| Funder | Grant reference number | Author |
| --- | --- | --- |
| National Institutes of Health | R35HL140014 | M Luisa Iruela-Arispe |
| National Institutes of Health | R01HL147187 | Casey E Romanoski |
| Foundation for the National Institutes of Health | FAPESP 2016/19968-3 | Vanessa Freitas |
| National Institutes of Health | K12 HD000849 | Yalda Afshar |
| National Institutes of Health | T32HL069766 | Yalda Afshar |
| Fondation Leducq | 21CVD03 | Miikka Vikkula |

The funders had no role in study design, data collection, and interpretation, or the decision to submit the work for publication.

### Author contributions

Yalda Afshar, Conceptualization, Data curation, Formal analysis, Funding acquisition, Validation, Investigation, Visualization, Methodology, Writing – original draft, Writing – review and editing; Feyiang Ma, Data curation, Formal analysis, Writing – review and editing; Austin Quach, Formal analysis, Writing – review and editing; Anhyo Jeong, Vanessa Freitas, Data curation, Investigation, Writing – review and editing; Hannah L Sunshine, Yasaman Jami-Alahmadi, Data curation, Writing – review and editing; Raphael Helaers, Miikka Vikkula, Resources, Software, Writing – review and editing; Xinmin

Li, Resources, Methodology, Writing – review and editing; Matteo Pellegrini, James A Wohlschlegel, Resources, Supervision, Writing – review and editing; Casey E Romanoski, Supervision, Investigation, Methodology, Writing – review and editing; M Luisa Iruela-Arispe, Conceptualization, Resources, Supervision, Funding acquisition, Investigation, Methodology, Writing – original draft, Project administration, Writing – review and editing

**Author ORCIDs**
Yalda Afshar http://orcid.org/0000-0003-3807-7022
Vanessa Freitas http://orcid.org/0000-0001-9613-8626
Yasaman Jami-Alahmadi http://orcid.org/0000-0001-8289-2222
Casey E Romanoski http://orcid.org/0000-0002-0149-225X
M Luisa Iruela-Arispe http://orcid.org/0000-0002-3050-4168

**Ethics**
Human umbilical cords were collected under Institutional Review Board (UCLA IRB#16-001694) at time of the delivery and processed 2–4 hr from time of birth. All samples were collected from participants who provided signed informed consent.

**Decision letter and Author response**
Decision letter https://doi.org/10.7554/eLife.81370.sa1
Author response https://doi.org/10.7554/eLife.81370.sa2

## Additional files

**Supplementary files**
• Supplementary file 1. RNAseq gene expression matrix by participant and environmental level conditions (cord, early culture, and late culture).

• Supplementary file 2. Liquid chromatography–mass spectrometry (LC–MS) protein expression by participant and environmental level conditions (cord and culture).

• Supplementary file 3. RNAseq gene comparison matrix by environmental condition (multi-tabulated excel): Tabs included ($n = 10$) are Cord versus Early, Cord versus Late, Cord versus Early/Late, Early versus Late, Culture versus 0.5 hr flow, Culture versus 8 hr flow, Culture versus 24 hr flow, Culture versus 48 hr flow, Flow (>8 hr) versus No Flow, Early flow versus No flow.

• Supplementary file 4. Biweight midcorrelation (bicor) between gene expression levels by condition and participant with module correlation (multi-tabulated excel): Tabs included ($n = 2$): all genes, non-coding genes.

• Supplementary file 5. Environmental driven gene expression and rescue with concordance and discordance (multi-tabulated excel): Table included ($n = 6$): cord, flow, up in both, down in both, up in cord, down in flow, down in cord, up in flow.

• Supplementary file 6. Single-cell RNA sequencing (scRNAseq) gene matrix of endothelial cell monolayer versus co-culture (MC vs. CC).

• Supplementary file 7. Single-cell RNA sequencing (scRNAseq) gene matrix of endothelial cell monolayer (MC) versus cord.

• Supplementary file 8. Single-cell RNA sequencing (scRNAseq) gene matrix of endothelial cell – smooth muscle cell co-culture (CC) versus cord.

• Supplementary file 9. Concordance of differentially expressed RNA and protein in cord versus culture.

• MDAR checklist

**Data availability**
All data generated or analyzed during this study are included in the manuscript and supporting file, Source Data files have been provided.

The following previously published datasets were used:

| Author(s) | Year | Dataset title | Dataset URL | Database and Identifier |
|---|---|---|---|---|
| McMahon SB, Stanek TJ | 2021 | The SAGA complex regulates early steps in transcription via its deubiquitylase module subunit USP22 | https://www.ncbi.nlm.nih.gov/geo/query/acc.cgi?acc=GSE158081 | NCBI Gene Expression Omnibus, GSE158081 |
| Gamba R | 2019 | CENP-C Cut&Run-seq | https://www.ncbi.nlm.nih.gov/geo/query/acc.cgi?acc=GSE156939 | NCBI Gene Expression Omnibus, GSE156939 |
| Lehtiö J, Vesterlund M | 2022 | Proteogenomic subtyping of chronic lymphocytic leukemia identifies subgroups with contrasting clinical outcome and distinct ex-vivo drug response profile | http://proteomecentral.proteomexchange.org/cgi/GetDataset?ID=PXD028936 | ProteomeXchange, PXD020958 |

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

# Appendix 1

## Appendix 1—key resources table

| Reagent type (species) or resource | Designation | Source or reference | Identifiers | Additional information |
|---|---|---|---|---|
| Biological sample (human) | Human umbilical vein endothelial cells | University of California, Los Angeles | N/A | Iruela-Arispe Lab and/or Afshar Lab |
| Biological sample (human) | Human umbilical vein tissue samples | University of California, Los Angeles | N/A | Iruela-Arispe Lab and/or Afshar Lab |
| Biological sample (human) | Human umbilical smooth muscle cells | University of California, Los Angeles | N/A | Iruela-Arispe Lab and/or Afshar Lab |
| Chemical compound, drug | MCDB-131 Complete Medium | VEC Technologies Inc | Cat# MCDB-131 WOFBS | |
| Chemical compound, drug | Dextran | Sigma-Aldrich | Cat# 31392-50G | |
| Chemical compound, drug | Fetal bovine serum | Omega Scientific, inc | Cat# FB-01; Lot# 871023 | |
| Chemical compound, drug | HBSS | Fisher Scientific | Cat# MT 21-023-CV | |
| Peptide, recombinant protein | Collagenase, Type 2 | Worthington Biochemical | Cat# LS004176 | |
| Chemical compound, drug | Trypsin | Fisher Scientific | Cat# MT-25-054CI | |
| Chemical compound, drug | eBioscience 1× RBC Lysis buffer | Thermo Fischer Scientific | Cat# 00433357 | |
| Peptide, recombinant protein | Pierce Bovine Serum Albumin | Thermo Fischer Scientific | Cat# 23209 | |
| Chemical compound, drug | Sodium deoxycholate | Sigma-Aldrich | Cat# D6750-25G | |
| Chemical compound, drug | 5% Mini-PROTEAN TBE Gel | Bio-Rad | Cat# 4565013 | |
| Peptide, recombinant protein | RNase-free DNase | QIAGEN | Cat# 79254 | |
| Chemical compound, drug | SuperSignal West Pico PLUS Chemiluminescent Substrate | Fisher Scientific | Cat# PI34580 | |
| Chemical compound, drug | SuperSignal West Femto Chemiluminescent Substrate | Fisher Scientific | Cat# PI34096 | |
| Chemical compound, drug | Restore Western Blot Stripping Buffer | Fisher Scientific | Cat# 21059 | |
| Chemical compound, drug | Ponceau S Solution, Bioreagent | Sigma | Cat# P7170 | |
| Chemical compound, drug | 4–20% Mini-PROTEAN TGX Precast Protein Gels, 12-well, 20 µl | Bio-Rad | Cat# 4561095 | |
| Chemical compound, drug | Precision Plus Protein Dual Color Standards | Bio-Rad | Cat# 1610374 | |
| Chemical compound, drug | Tween 20 | Fisher Scientific | Cat# BP337500 CAS 9005-64-5 | |
| Chemical compound, drug | Sodium Orthovanadate, >99% | Thermo Fisher Scientific | Cat# AC205330500 | |
| Chemical compound, drug | Complete, EDTA-Free Protease Inhibitor Cocktail | Sigma | Cat# 11873580001 | |
| Chemical compound, drug | Triton X-100 | Fisher | Cat# BP151 | |

*Appendix 1 Continued on next page*

*Appendix 1 Continued*

| Reagent type (species) or resource | Designation | Source or reference | Identifiers | Additional information |
|---|---|---|---|---|
| Chemical compound, drug | Sodium chloride | Fisher | Cat# S271 | |
| Chemical compound, drug | Tris–HCl | Fisher | Cat# BP153 | |
| Chemical compound, drug | Tris-Base | Fisher | Cat# BP152 | |
| Chemical compound, drug | Sodium dodecyl sulfate | Fisher | Cat# BP166 | |
| Chemical compound, drug | Glycine | Dot Scientific | Cat# DSG36050 | |
| Chemical compound, drug | Bromophenol Blue | Fisher | Cat# B392 | |
| Chemical compound, drug | 2 Mercaptoethanol, 99%, extra pure | Acros Organics | Cat# 125472500 | |
| Chemical compound, drug | Dulbecco's Modified Eagle Medium (DMEM) with L-Glutamine and 4.5 g/L Glucose; Without Sodium Pyruvate | Corning | Cat# 10017CV | |
| Chemical compound, drug | Glycerol | Invitrogen | Cat# 15514 | |
| Commercial assay, kit | RNeasy Plus Micro Kit | QIAGEN | Cat# 74034 | |
| Commercial assay, kit | Ribo-Zero rRNA removal kit | Ilumina | Cat# MRZH11124 | |
| Commercial assay, kit | Nextera Index kit | Ilumina | Cat# FC-121-1011 | |
| Commercial assay, kit | MinElute PCR Purification Kit | QIAGEN | Cat# 28004 | |
| Commercial assay, kit | QIAquick PCR Purification Kit | QIAGEN | Cat# 28104 | |
| Commercial assay, kit | Nextera DNA Sample Preparation Kit | Ilumina | Cat# FC-121-1030 | |
| Chemical compound, drug | NEBNext High-Fidelty 2× PCR Master Mix | New England Biolab | Cat# MO541S | |
| Chemical compound, drug | SYBR Green I Nucleic Acid Gel Stain | Fisher Scientific | Cat# S7563 | |
| Commercial assay, kit | ChIP DNA Clean & Concentrator | Zymo | Cat# D5205 | |
| Commercial assay, kit | 10× reagents for library | 10xGenomics | Cat# 1000075 | |
| Commercial assay, kit | CD31 MicroBead Kit, human | Miltenyi Biotec | Cat# 130-091-935 | |
| Commercial assay, kit | Trans-Blot Turbo RTA Midi Nitrocellulose Transfer Kit | Bio-Rad | Cat# 1704271 | |
| Commercial assay, kit | Thermo Scientific Pierce Detergent Compatible Bradford Assay | Fisher Scientific | Cat# PI23246 | |
| Commercial assay, kit | QuadroMACS Starting Kit (LS) | Miltenyi Biotec | Cat# 130-091-051 | |
| Other (deposited data) | Raw data files for bulk RNAseq | NCBI GEO | GSE158081 | https://www.ncbi.nlm.nih.gov/gds |
| Other (deposited data) | Raw data files for scRNAseq | NCBI GEO | GSE156939 | https://www.ncbi.nlm.nih.gov/gds |
| Other (deposited data) | Raw data files for LC/MS | Proteome Xchange Consortium PRIDE | PXD020958 | https://www.proteomexchange.org/ |
| Software, algorithms | STAR (2.5.4a) | *Dobin et al., 2013* | https://github.com/alexdobin/STAR | |
| Software, algorithms | FeatureCounts | *Liao et al., 2014* | http://subread.sourceforge.net/ | |
| Software, algorithms | Bioconductor package DESeq2 | *Love et al., 2014* | https://bioconductor.org/packages/release/bioc/html/DESeq2.html | |

*Appendix 1 Continued on next page*

*Appendix 1 Continued*

| Reagent type (species) or resource | Designation | Source or reference | Identifiers | Additional information |
|---|---|---|---|---|
| Software, algorithms | Heatmap.2 In R package | Online | https://www.rdocumentation.org/packages/gplots/versions/3.0.4/topics/heatmap.2 | |
| Software, algorithms | 10× Chromium Single Cell Software Loupe Browser (visualization tools), version 4.1 | Online | https://support.10xgenomics.com/single-cell-gene-expression/software/overview/welcome | |
| Software, algorithms | Image Lab, Version 6.0.0.0 build 25 | Bio-Rad Laboratories | | |

