## [Editor Report]

The findings of this study, focused on endothelial cells, are fundamental and of broad interest to various fields using cultured primary cells. The authors provide compelling evidence of how the culture conditions impact on gene expression.

---

## [Decision Letter]

**Decision letter after peer review:**

Thank you for submitting your article "Transcriptional drifts associated with environmental changes in endothelial cells" for consideration by *eLife*. Your article has been reviewed by 3 peer reviewers, and the evaluation has been overseen by a Reviewing Editor and Edward Morrisey as the Senior Editor. The following individuals involved in the review of your submission have agreed to reveal their identity: Mete Civelek (Reviewer #1); Zhen Bouman Chen (Reviewer #2).

Essential revisions:

We list here two suggestion points that we think are essential to support the claims made. These are:

1) Clear description of experimental and analytical details with a particular focus on the comparison analysis.

2) Determine the presence of other cells in primary cultures in order to establish the level of non-endothelial cells in the cultures that may influence responses.

*Reviewer #2 (Recommendations for the authors):*

1. There are multiple comparisons made in the study. Please include more details as follows:

a. Please indicate the cutoffs used to define DEGs in different comparisons.

b. In Figure 1 e-g, it was indicated that "we considered only differences between cord and culture signatures going forward". Was early or late culture used from here on, or both?

c. What passage of ECs were used for the flow and co-culture experiment? Please indicate in the figure legends accordingly.

d. For the 17% of genes rescued by flow and 9% by co-culture with VSMC, what does "rescue" mean exactly? Does it mean partially or completely to "cord" condition?

e. How are the donors used for the flow and co-culture experiment selected among the 7 donors used initially?

2. eNOS is a hallmark gene of ECs and is flow-inducible. Its expression in various datasets should be shown to facilitate the comprehension of the genome-wide data by the readers. On the other hand, VCAM1 and ICAM1, which are also flow-regulated, should also be included.

3. Are all the genes analyzed in the current study protein-coding? Have the authors analyzed non-coding genes and is there expression contributing to the transcriptomic difference under different conditions? Can the authors include some example lncRNAs that have been shown to be flow-responsive, e.g. LINC00520, RAMP2-AS1, STEEL, and SENCR?

4. Complementary to the gene expression data, it would be great if authors include representative imaging data to show the morphology of freshly isolated, early and late passage, and flow-imposed ECs, even just for one cord ECs.

5. The data collected from RNA and protein profiling is excellent. Can authors include in the supplemental data the list of genes showing a high level of correlation (e.g. coefficient of r>0.8) between RNA and protein levels?

6. Figure 3A, how did the author conclude that the transcriptional profile of cultured cells under flow approximates (to) better to the in vivo transcriptome when compared to static states? It appears that the 48 hr flow is in the middle on PC1. How many data points are used as "flow" as there are three different time points?

7. While the data in Figure 3 is very interesting, the writing is rather difficult to understand. Please revise Lines 392-410 to facilitate the comprehension of the results. Moreover, please improve the labels of Figure 3C and D, which are difficult to follow.

8. It is noted that early and late culture overlap in 93% of the genes. It is curious what the other 7% of genes are. Are they related to cell senescence and cell cycle regulation? Are these genes significantly differentially expressed compared to the Cord state and are they regulated by shear stress and/or VSMC co-culture?

9. Please provide supplemental tables showing genes rescued by flow and those rescued by co-culture (maybe described in detail), and discuss the potential synergistic effects of these two environmental cues. This will make the study more complete.

10. Figure 4 legend: please check 4E and F. G and H are missing legends. Figure 5F, please complete the legend. (Please check figure legends throughout to ensure the correctness and completeness and the inclusion of adequate details.)

11. On Lines 418-421, it is indicated triplicates were profiled, and "endothelial and smooth muscle cells were isolated from the same cord". However, in Table 1, there appear two different donors. Please clarify.

---

## [Author Response]

Essential revisions:We list here two suggestion points that we think are essential to support the claims made. These are:1) Clear description of experimental and analytical details with a particular focus on the comparison analysis.

The revised manuscript now includes an expanded description of both the experimental details and analytical evaluation in the Methods and figure legends. See specifically, cell culture, RNA isolation, shear stress application, and Supplemental Figure 1. Additionally, figure legends have clarified the comparisons.

2) Determine the presence of other cells in primary cultures in order to establish the level of non-endothelial cells in the cultures that may influence responses.

We appreciate the concern raised. The revised version of the manuscript now clearly states the approaches that assess any level of contamination in the culture. These included: scRNAseq from the isolated cord cells (Supplemental Figure 1C) and FACs of P2 cultured cells. The scRNAseq in particular, shows data from three independent cord isolations. As can be appreciated from that panel, the only cells contaminating the isolates are blood cells (red blood cells and CD45+ cells) which are no longer present at P2 cultures (as all cells present are double positive for CD31 and VE-Cadherin).

In terms of data analysis, in Supplemental Figure 5, we show more scRNAseq details, including the process by which we filter the low-quality cells. In the filtering steps, we removed cluster 9-13 because cluster 9, 10, 12, and 13 had low number of genes detected, representing low-quality cells. Cluster 11 had high level of interferon induced genes expressed, representing stressed cells in the library, and this was noted in the text. Transcriptional marker analysis in the scRNAseq revealed that the primary cultures used bonefide endothelial (when isolated for this cell type) or smooth muscle cells (when isolated for this cell type). No other cell types were found in the primary cultures.

Reviewer #2 (Recommendations for the authors):1. There are multiple comparisons made in the study. Please include more details as follows:a. Please indicate the cutoffs used to define DEGs in different comparisons.

Thank you for the comment, the FDR < 0.05 cutoff is now clearly defined in the method description.

b. In Figure 1 e-g, it was indicated that "we considered only differences between cord and culture signatures going forward". Was early or late culture used from here on, or both?

Since no significant differences were observed between early and late all other comparisons were made with early cultures. This has now been clearly noted in the study results.

c. What passage of ECs were used for the flow and co-culture experiment? Please indicate in the figure legends accordingly.

Thank you. We have clarified in the results (see point b) that we used early passage (P2-3) for every subsequent experiment for all cell culture.

d. For the 17% of genes rescued by flow and 9% by co-culture with VSMC, what does "rescue" mean exactly? Does it mean partially or completely to "cord" condition?

Thank you for this point. What we mean by rescue is to reach expression similar to that in the cord condition. This has been now clarified in the text and Supplemental Excel 5-8 are available to provide gene level data of the rescue in relation to the cord.

e. How are the donors used for the flow and co-culture experiment selected among the 7 donors used initially?

We have clarified this in the Table 1 and the Methods. The donors for the flow and co-culture experiments are different in each experiment and the same donor does not undergo shear stress and or co-culture. The expansion in Table 1 aims to clarify this.

2. eNOS is a hallmark gene of ECs and is flow-inducible. Its expression in various datasets should be shown to facilitate the comprehension of the genome-wide data by the readers. On the other hand, VCAM1 and ICAM1, which are also flow-regulated, should also be included.

We appreciate the comment and have now included a new panel figure (Supplemental Figure 3D) that compares well-known flow-responsive genes (NOS3, VCAM1, ICAM1, KLF2, KLF4) along with others identified by this dataset. As it can be noted by the heatmap, there is a clear increase in known flow responsive genes, consistent with what has been shown in the literature.

Furthermore, “to facilitate the comprehension of the genome-wide data by the readers”, we have also developed a user-friendly electronic platform that will allow readers to inquire about the profile of any gene of interest (Flow Profiler, included in supplementary figure 5).

3. Are all the genes analyzed in the current study protein-coding? Have the authors analyzed non-coding genes and is there expression contributing to the transcriptomic difference under different conditions? Can the authors include some example lncRNAs that have been shown to be flow-responsive, e.g. LINC00520, RAMP2-AS1, STEEL, and SENCR?

This is an excellent point. In this study, we analyzed both coding and noncoding transcripts. Of the 5 flow-responsive noncoding RNA listed only SENCR was found in the expression dataset. The non-coding genes were included in the original results but the type was not explicitly stated. In the revised version of Supplementary file 4, we have added the "genotype” column to the association table and included a separate tab with only the non-coding transcripts.

4. Complementary to the gene expression data, it would be great if authors include representative imaging data to show the morphology of freshly isolated, early and late passage, and flow-imposed ECs, even just for one cord ECs.

Great suggestion. We have now included a new Supplemental Figure 1 (and all Supplemental Figures have been re-numbered) to include cell morphology of the in vivo and in vitro endothelial cells, the smooth muscle cells for the co-culture experiments, as well as the cells under flow. Thank you for this suggestion. We believe it will be helpful for the readers to see the phenotype. In addition, we have now included extensive quality control information (including scRNAseq and FACs) that speaks to the quality of the isolations performed.

5. The data collected from RNA and protein profiling is excellent. Can authors include in the supplemental data the list of genes showing a high level of correlation (e.g. coefficient of r>0.8) between RNA and protein levels?

Prompted by the reviewer, we have done now more data mining and included an additional supplementary spreadsheet comparing early culture associations between RNA and protein levels. Perhaps expectedly, the most significant protein changes coincide with significant corresponding changes in RNA but not the most significant changes in RNA did not coincide with significant changes in corresponding protein levels. These findings have been included as an additional Supplemental Excel 9.

6. Figure 3A, how did the author conclude that the transcriptional profile of cultured cells under flow approximates (to) better to the in vivo transcriptome when compared to static states? It appears that the 48 hr flow is in the middle on PC1. How many data points are used as "flow" as there are three different time points?

Based on the transcriptional similarity of the different flow time points (Figure 2B) we consolidated the “under flow” label for Figure 3A for clarity. Based on the polarity of PC1 (from left to right: cord > extended flow > early culture + short-term flow > culture) and its magnitude (31% of the covariance in the dataset), we concluded that PC1 primarily represents the differences between Cord and Culture samples (we interpret the PC2 to represent differences between short-term and extended flow). Since the extended flow samples are in the middle position between cord and late culture, we interpret this as a partial rescue of the differences imparted by culture. This interpretation is now included in the revised manuscript and the figure legend.

7. While the data in Figure 3 is very interesting, the writing is rather difficult to understand. Please revise Lines 392-410 to facilitate the comprehension of the results. Moreover, please improve the labels of Figure 3C and D, which are difficult to follow.

We appreciate bringing this to our attention. We have revised the text in the results for purpose of clarity and have also improved and expanded on the legends for Figures 3C and D.

8. It is noted that early and late culture overlap in 93% of the genes. It is curious what the other 7% of genes are. Are they related to cell senescence and cell cycle regulation? Are these genes significantly differentially expressed compared to the Cord state and are they regulated by shear stress and/or VSMC co-culture?

Excellent point, we have now mined the data and clarify the GOs for the 7% genes and further identified them, as well as conducted additional analyses to compare them to cord, shear stress and co-culture conditions. This information, as well as the comparisons, is included in Supplementary file 4.

9. Please provide supplemental tables showing genes rescued by flow and those rescued by co-culture (maybe described in detail), and discuss the potential synergistic effects of these two environmental cues. This will make the study more complete.

We appreciate this request and believe it has strengthened the utility of our dataset and improved the manuscript. The information is now provided in an updated Supplementary files 4-5 and we have extended the discussion to include this rescue phenomena.

10. Figure 4 legend: please check 4E and F. G and H are missing legends. Figure 5F, please complete the legend. (Please check figure legends throughout to ensure the correctness and completeness and the inclusion of adequate details.)

Thank you. This has been corrected and expanded upon for both experimental detail and clarity.

11. On Lines 418-421, it is indicated triplicates were profiled, and "endothelial and smooth muscle cells were isolated from the same cord". However, in Table 1, there appear two different donors. Please clarify.

Thank you for catching this lack of clarity. Technical replicates were run and analyzed of the same two donors as described in Table 1 and in the Figure and Supplemental Figure legends. We have clarified this text and language in the Methods, Results and legends.